# State Regularized Policy Optimization
# on Data with Dynamics Shift

**Zhenghai Xue**[1]    **Qingpeng Cai**[2]    **Shuchang Liu**[2]    **Dong Zheng**[2]
**Peng Jiang**[2]    **Kun Gai**[3]    **Bo An**[1]
[1]Nanyang Technology University, Singapore
[2]Kuaishou Technology    [3] Unaffiliated
zhenghai001@e.ntu.edu.sg  boan@ntu.edu.sg  gai.kun@qq.com
{caiqingpeng,liushuchang,zhengdong,jiangpeng}@kuaishou.com

## Abstract

In many real-world scenarios, Reinforcement Learning (RL) algorithms are trained
on data with dynamics shift, i.e., with different underlying environment dynamics.
A majority of current methods address such issue by training context encoders to
identify environment parameters. Data with dynamics shift are separated according
to their environment parameters to train the corresponding policy. However, these
methods can be sample inefficient as data are used *ad hoc*, and policies trained for
one dynamics cannot benefit from data collected in all other environments with
different dynamics. In this paper, we find that in many environments with similar
structures and different dynamics, optimal policies have similar stationary state
distributions. We exploit such property and learn the stationary state distribution
from data with dynamics shift for efficient data reuse. Such distribution is used
to regularize the policy trained in a new environment, leading to the SRPO (**S**tate
**R**egularized **P**olicy **O**ptimization) algorithm. To conduct theoretical analyses,
the intuition of similar environment structures is characterized by the notion of
homomorphous MDPs. We then demonstrate a lower-bound performance guarantee
on policies regularized by the stationary state distribution. In practice, SRPO can be
an add-on module to context-based algorithms in both online and offline RL settings.
Experimental results show that SRPO can make several context-based algorithms
far more data efficient and significantly improve their overall performance.

## 1 Introduction

Reinforcement Learning (RL) has achieved great success in solving challenging sequential decision-
making problems [1, 2]. Unfortunately, existing RL methods usually assume that agents are trained
and evaluated in exactly the same environment, which is often not the case in real-world applications
where environment dynamics can vary a lot. For example, the recommendation engine of social
apps may need to deal with time-varying and heterogeneous user preferences [3, 4]. A robot arm
may operate in different scenarios with different joint frictions and medium densities [5]. In these
cases, the agent has to work with the trajectory data from *different* environment dynamics, i.e., data
with *dynamics shift*, which will bias the learning process and lead to poor performance. In fact,
some empirical studies [6, 5] demonstrate that general RL algorithms [7, 8] can easily be misled by
different environment dynamics and fail to train a good policy.

In recent years, considerable research efforts have been devoted to addressing the dynamics shift and
learning generalizable policies for environments with changing dynamics. One common practice is
to train a context encoder [9, 10, 11] to associate the environment dynamics with a latent variable.
The policy is then trained with the latent variable as an additional input [12]. One issue with this

37th Conference on Neural Information Processing Systems (NeurIPS 2023).

practice is that policies conditioned on a specific latent variable can only learn from data collected in the environment corresponding to that latent variable. In other words, data with different dynamics are used in an *ad hoc* manner. The generalizability of context encoders relies on the expressive power of neural networks. However, neural networks are prone to overfit and behave poorly when extrapolating. As an example, we benchmarked CaDM [9], which is one of the context-based algorithms, under Ant environments with different gravities and display the results in Fig. 1. Although it can outperform PPO [7] due to its adaptability from context encoders, CaDM fails to constantly improve its performance with more data from different environment dynamics. To mitigate the problem of inefficient data use, there are some attempts that leverage Importance Sampling (IS) [13, 5, 14]. Given the dynamics of the target environment, samples from the source environments are assigned with larger importance weights if they are more likely to happen in the target environment and vice versa. Compared to training context

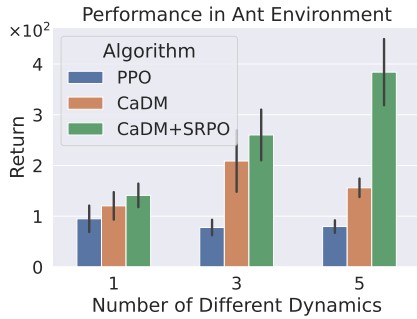

Figure 1: Performance comparison of PPO [7], CaDM [9] and CaDM+SRPO in the Ant environment, where SRPO is our proposed state regularized policy optimization method. Details of the experiment setup are in Sec. 5.1.

encoders, IS-based methods manage to proactively *exploit* the data from other dynamics. However, such methods require prior knowledge about the dynamics of the target environment. Also, it is notoriously hard to balance the bias and variance when calculating the IS weights.

This paper proposes a new RL paradigm that can explicitly leverage data with dynamics shift. It is also free of the aforementioned drawbacks of IS-based methods. We find that the stationary state distribution induced by optimal policies (later termed optimal state distribution) is similar across a set of environments with similar structures and different environment dynamics. For example, given heterogeneous preferences of users, a video recommendation system may choose different videos to recommend, but the optimal states are the same: users keep pressing the "like" or "save" button and continue watching for a long time. More concretely, the optimal state distribution in one environment dynamics can be informative for training policies in all other different dynamics. We therefore propose a constrained policy optimization (CPO) [15] formulation that requires the policy not only to optimize the cumulative return, but also to generate a stationary state distribution close to the optimal state distribution. By relating optimality to high-reward states [16], we are able to approximate the optimal state distribution from trajectory data regardless of the underlying dynamics, providing a unified and efficient approach to exploiting these data.

Summarizing these ideas, we propose the SRPO (**S**tate **R**egularized **P**olicy **O**ptimization) algorithm. SRPO works as an add-on module in both online and offline context-based RL algorithms such as CaDM [9] and MAPLE [12] to increase their sample efficiency, leading to the CaDM+SRPO and MAPLE+SRPO algorithms. We provide a lower-bound performance guarantee on policies in one dynamics regularized by the optimal state distribution in other dynamics. This theoretically demonstrates the effectiveness of the SRPO algorithm in using data with dynamics shift. Empirical results in both online and offline settings show that SRPO can significantly improve both the data efficiency and the overall performance of several state-of-the-art context-based RL algorithms. We also perform ablation studies to demonstrate the effectiveness of each component in the SRPO algorithm.

## 2 Backgroud

### 2.1 Preliminaries

A Markov Decision Process (MDP) can be defined by a tuple $(\mathcal{S}, \mathcal{A}, T, r, \gamma, \rho_0)$, where $\mathcal{S}$ is the state space, $\mathcal{A}$ is the bounded action space with actions $a \in (-1, 1)$, $T(s'|s, a) \in [0, 1]$ and $r(s, a, s') \in [-R_{\max}, R_{\max}]$ are the transition and reward functions. $\gamma \in (0, 1)$ is the discount factor and $\rho_0(s)$ is the initial state distribution. In MDPs with deterministic transitions, we denote $T(s, a)$ as the transition function with a slight abuse of notation, and $(T, \varepsilon)$ as $\{T' \mid |T(s, a) - T'(s, a)| < \varepsilon, \ \forall s \in \mathcal{S}, a \in \mathcal{A}\}$ which is the $\varepsilon-$neighbourhood of $T$. RL aims at maximizing the accumulated return of policy $\pi$:

$\eta_T(\pi) = E_{\pi,T}\left[\sum_{t=0}^{\infty}\gamma^t r(s_t, a_t)\right]$, where the expectation is computed with $s_0 \sim \rho_0$, $a_t \sim \pi(\cdot|s_t)$, and $s_{t+1} \sim T(\cdot|s_t, a_t)$. The optimal policy $\pi^*$ is defined as $\pi_T^* = \arg\max_{\pi} \eta_T(\pi)$. In an MDP with a policy $\pi$, the Q-value $Q_T^\pi(s, a)$ denotes the expected return after taking action $a$ at state $s$: $Q_T^\pi(s, a) = E_{\pi,T}\left[\sum_{t=0}^{\infty}\gamma^t r(s_t, a_t)|s_0 = s, a_0 = a\right]$. The value function is defined as $V_T^\pi(s) = \mathbb{E}_{a\sim\pi(\cdot|s)}Q_T^\pi(s, a)$ with $V_T^*(s)$ being the shorthand for $V_T^{\pi_T^*}(s)$. It satisfies the optimal Bellman Equation $V_T^*(s) = \max_a r(s, a) + \gamma\mathbb{E}_{s'\sim T(\cdot|s,a)}V_T^*(s')$. We can also define the stationary state distribution (also known as state occupation function) as $d_T^\pi(s) := (1 - \gamma)\sum_{t=0}^{\infty}\gamma^t P_T(s_t = s \mid \pi)$ with $d_T^*(s)$ being the shorthand for $d_T^{\pi_T^*}(s)$.

The Hidden Parameter Markov Decision Process (HiP-MDP) captures a class of MDPs with different transition functions and the same reward function by introducing a set of hidden parameters. Specifically, an HiP-MDP is defined by a tuple $(\mathcal{S}, \mathcal{A}, \Theta, T, r, \gamma, \rho_0)$, where $\Theta$ is the space of hidden parameters. The transition function $T_\theta(s'|s, a, \theta)$ is parameterized not only by states and actions, but also by a hidden parameter $\theta$ sampled from $\Theta$. The action gap of an HiP-MDP is defined as $\Delta = \min_{\theta\in\Theta}\min_{s\in\mathcal{S}}\min_{a\neq\pi^*(s)} V_{T_\theta}^*(s) - Q_{T_\theta}^*(s, a)$, which reflects the minimum gap between an optimal action and all other sub-optimal actions.

## 2.2 Related Work

**MDPs with Different Dynamics**  The setting of HiP-MDP [17] was proposed to model a set of variations in the environment dynamics. The problem is intensively investigated in recent years [18] and these researches fall into three categories, i.e., encoder-based, Important Sampling (IS)-based and meta-RL based algorithms. Encoder-based methods extract the hidden parameters from trajectories with variational inference [19] or auxiliary loss [9]. These hidden parameters are used as inputs to the transition function [9] or policy network [20, 12]. Unfortunately, these methods train dynamics-specific policies from the trajectory data of each hidden parameter independently, which leads to poor sample efficiency. Instead, our method uses the data from all dynamics to learn an optimal state distribution that facilitates the policy learning. IS-based methods compute the importance ratio between transition probabilities under different dynamics and modify the replay buffer [13, 14, 5] according to the transition probabilities in the test environments, which is often not available in real-world scenarios. Finally, meta-RL algorithms [21, 22] can adapt to environments with new dynamics through fine-tuning on a small amount of data from the test environment. In contrast, our method can be directly applied to new environment dynamics by making a zero-shot transfer.

**Behavior Regularized Methods**  The idea of constrained policy optimization (CPO) [15] is widely used in RL. Most researches focus on behavior regularized methods, i.e., adding policy constraints based on another policy distribution, as shown in the following optimization problem:

$$\max_{\pi} \mathbb{E}_{s,a\sim\mathcal{D}}\left[\mathbb{E}_{a'\sim\pi(\cdot|s)}Q(s, a')\right] \qquad \text{s.t.} \quad \mathbb{E}_{s\sim\mathcal{D}}\left[\hat{D}(\pi(\cdot|s)\|\hat{\pi}(\cdot|s))\right] < \varepsilon, \qquad (1)$$

where $\mathcal{D}$ is the replay buffer, $\hat{\pi}$ is the regularizing policy, and $\hat{D}$ is a certain distance measure. Maximum-Entropy RL [23, 8] can be considered as CPO with a uniform policy distribution. The sparse action tasks [24] can be solved by CPO with a sparse policy distribution. Besides, many offline RL algorithms [25, 26, 27] are based on the idea of constraining the current policy distribution to be close to the dataset's policy distribution. However, data sampled from environments with different dynamics can have distinct optimal policies, as illustrated in Sec. 3.1. In such cases, $\hat{\pi}$ in Eq. (1) may include policies that do not match the current environment, and can therefore be misleading. So *behavior regularization* in Eq. (1) would fail on data with dynamics shift. Differently, our proposed method is based on *state regularization*, which is more suitable when learning from data with dynamics shift.

**Leveraging stationary state distributions**  The stationary state distribution $d_T^\pi(s)$ of policy $\pi$ and dynamics $T$ is an important feature that can measure the differences in policies and transition functions. It has already been exploited in many researches. In Off-Policy RL, Islam et. al [28] estimates the stationary state distributions of both the current policy and the mixed buffer policy. It

then computes the off-policy policy gradient with the constraint that the two distributions should be close. Some Off-Policy Policy Evaluation (OPE) algorithms [29, 30] use the steady-state property of stationary Markov processes to estimate the stationary state distributions. In Imitation Learning (IL), state-only IL algorithms [31, 32] requires the stationary state distribution of the current policy to be close to that of the expert policy. In Inverse RL (IRL), [33] learns a stationary reward function by computing the gradient of the distance between agent and expert state distribution w.r.t. reward parameters. In Offline RL, [34] requires the stationary state distribution of the learning policy and the behavior policy to be close and perform conservative updates. The use of such distributions in our paper is similar to some researches on sim-to-real [35, 36]. They propose to match the next state distribution in the imperfect simulator and the real environment with inverse dynamics model. They implicitly relies on the idea that the same state distribution should generate similar returns in environments with different dynamics. We formulate the idea in this paper with theorems and quantitatively analyse such similarity in various conditions.

## 3   State Regularized Policy Optimization

In this section, we first give motivating examples on why the optimal state distribution in one environment dynamics can be informative in all other different dynamics. A constrained policy optimization formulation is then proposed in Sec. 3.2 based on the optimal state distribution. Solving this optimization problem gives rise to our State Regularized Policy Optimization (SRPO) algorithm that can leverage data with dynamics shift to improve the policy performance.

### 3.1   Motivating Example

The key intuition behind SRPO is that the optimal state distribution is similar across environments. Consider an example of the Inverted Pendulum environment in Fig. 2. We train two policies in the environments with gravities of 5 and 10 until convergence. Then the kernel density estimation [37] technique is employed to estimate the state and action density of the data collected by the two policies in different areas. It can be observed from the figure that collected data have the same high state density region with low pendulum speed and small pendulum angle, while the action distribution has different density peaks. It demonstrates that the state distribution of data generated by the optimal policy can be similar regardless of the environment dynamics, and therefore can serve as a reference distribution to regularize the training policy in environments with new dynamics. More demonstrating examples can be found at Appendix B.3.

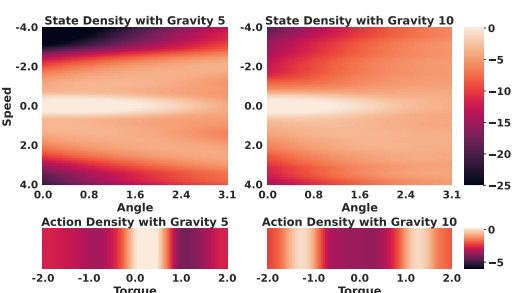

Figure 2: Visualization of state and action densities in data sampled from the Inverted Pendulum environment with gravity 5 and 10. Under both gravities, the state distribution has high density with low pendulum speed and small pendulum angle. Meanwhile, the action distribution has different peaks in density under different gravities.

### 3.2   State Regularized Policy Optimization

Based on the intuition of informative optimal state distribution, we develop a novel technique that regulates RL algorithms to generate a stationary state distribution that is close to the optimal one. Specifically, we propose the following constrained policy optimization formulation:

$$\max_{\pi} \; \mathbb{E}_{s_t,a_t \sim \tau_\pi} \left[ \sum_{t=0}^{\infty} \gamma^t r\left(s_t, a_t\right) \right] \qquad \text{s.t.} \quad D_{\mathrm{KL}}\left(d_\pi(\cdot) \| \zeta(\cdot)\right) < \varepsilon, \tag{2}$$

where $\zeta(s)$ is the optimal state distribution in other environment dynamics. By introducing the stationary state distribution, the optimization problem defined in Eq. (2) extends the regularization of in-distribution data to data with distribution shift[1].

We solve Eq. (2) by casting it to the following unconstrained optimization problem via Lagrange multipliers[2]:

$$L = -\mathbb{E}_{s_t,a_t \sim \tau} \left[ \sum_{t=0}^{\infty} \gamma^t \left( r(s_t, a_t) + \lambda \log \frac{\zeta(s_t)}{d_\pi(s_t)} \right) \right] - \frac{\lambda \varepsilon}{1 - \gamma}, \tag{3}$$

where $\lambda > 0$ is the Lagrangian Multiplier. It is noteworthy that in addition to the multiplier term, the only difference of Eq. (3) and the reward-maximization objective of RL is that the logarithm of probability density ratio $\lambda \log \frac{\zeta(s_t)}{d_\pi(s_t)}$ is added to the reward term $r(s_t, a_t)$. Therefore, one can easily apply our scheme to a wide range of RL algorithms by augmenting the reward function with the density ratio.

### 3.3 Data-based Surrogate of the Density Ratio

The main challenge in solving Eq. (3) is to compute the density ratio $\frac{\zeta(s)}{d_\pi(s)}$ because obtaining the optimal state distribution during online training or given suboptimal offline dataset is infeasible. Also, $d_\pi(s)$ is intractable if the state space is continuous. Motivated by recent advances in adversarial training [38, 39], we propose a sample-based surrogate for the density ratio $\frac{\zeta(s)}{d_\pi(s)}$.

**Proposition 3.1.** *In a GAN, when the real data distribution is $\zeta(s)$ and the generated data distribution is $d_\pi(s)$, the output of the discriminator $D(s)$ follows*

$$\frac{D(s)}{1 - D(s)} = \frac{\zeta(s)}{d_\pi(s)}. \tag{4}$$

We discuss the relation of this sample-based surrogate with f-divergences and Off-Policy RL in Appendix A.3. To train the discriminator $D(s)$, we need to generate samples that is close to the optimal state distribution $\zeta(s)$ and away from $d_\pi(s)$, which is sort of the average state distribution. Motivated by [16], we model state optimality by a variable $\mathcal{O}_t$. As shown in Fig. 3, we regard the state $s_t$ in MDP as a hidden state in a Hidden Markov Model (HMM), and introduce the binary observation state $\mathcal{O}_t$. $\mathcal{O}_t = 1$ denotes that $s_t$ is the optimal state at timestep $t$. The observation model is given by

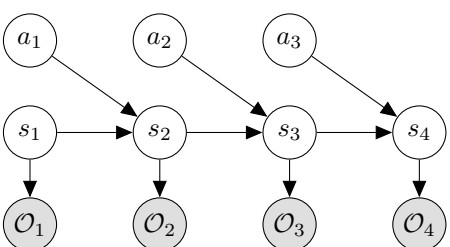

Figure 3: HMM in MDP with optimality variables $\mathcal{O}_t$.

$$p(\mathcal{O}_t|s_t) = \max_{a_t} \exp[\gamma^t (r(s_t, a_t) - R_{\max})]. \tag{5}$$

We can therefore compute the state density ratio $\frac{\zeta(s)}{d_\pi(s)}$ as

$$\frac{\zeta(s)}{d_\pi(s)} = \frac{d_\pi(s|\mathcal{O}_{0:\infty})}{d_\pi(s)} = \frac{p(\mathcal{O}_{0:\infty}|s,\pi)d_\pi(s)}{p(\mathcal{O}_{0:\infty}|\pi)d_\pi(s)} = \frac{\mathbb{E}_t[p(\mathcal{O}_{0:t-1}|s_t,\pi)p(\mathcal{O}_{t:\infty}|s_t,\pi)]}{p(\mathcal{O}_{0:\infty}|\pi)}, \tag{6}$$

where the second equation follows the Bayes' law. The last term is related to the forward probability $\alpha_t(s_t) = p(\mathcal{O}_{0:t-1}|s_t, \pi)$ and backward probability $\beta_t(s_t) = p(\mathcal{O}_{t:\infty}|s_t, \pi)$ in the HMM. We discuss in Appendix A.2 that $\beta_t(s_t)$ is positively related to a soft version of MDP's state value $V_\pi(s_t)$. Also, $\alpha_t(s_t)$ makes a little influence on the overall density ratio. Therefore, the input $s$ will be more likely to be sampled from distribution $\zeta(s)$ if it has a higher state value $V(s)$ than average. With this idea, we are able to build training samples for the discriminator $D(s)$.

---

[1]A similar form of Eq. (2) (See Eq. 1 in Sec. 2.2) is employed in Offline RL algorithms to ensure conservative policy updates. But it restricts the training data to be sampled from the same environment.

[2]The detailed derivations of the Lagrangian can be found in Appendix A.1.

**Algorithm 1** The workflow of SRPO on top of MAPLE [12].

---
1: **Input:** $\phi_\varphi$ as a context encoder parameterized by $\varphi$; Adaptable policy network $\pi_\theta$ parameterized by $\theta$; Adaptable value network $V_\psi$ parameterized by $\psi$; Offline dataset $\mathcal{D}_{\text{off}}$; Rollout horizon $H$; State partition ratio $\rho$; State discriminator $D_\delta$ parameterized by $\delta$; Regularization coefficient $\lambda$.
2: **for** 1, 2, 3, . . . **do**
3:     **for** $t = 1, 2, \ldots, H$ **do**
4:         Sample $z_t$ from $\phi_\varphi(z \mid s_t, a_{t-1}, z_{t-1})$ and then sample $a_t$ from $\pi_\theta(a \mid s_t, z_t)$.
5:         Rollout and get transition data $(s_{t+1}, r_t, d_{t+1}, s_t, a_t, z_t)$. Then add it to $\mathcal{D}_{\text{rollout}}$.
6:     **end for**
7:     Update the context encoder $\phi_\varphi$ according to MAPLE.
8:     Sample a batch $\mathcal{D}_{\text{batch}}$ from $\mathcal{D}_{\text{off}}$ and $\mathcal{D}_{\text{rollout}}$ and rank them by their state-values estimated by $V_\psi$; Add $\rho|\mathcal{D}_{\text{batch}}|$ states with higher state-values to $\mathcal{D}_{\text{real}}$ and the others to $\mathcal{D}_{\text{fake}}$.
9:     Train the discriminator $D_\delta$ with nll loss.
10:     For one-step transition $(s_{t+1}, r_t, d_{t+1}, s_t, a_t, z_t)$ in $\mathcal{D}_{\text{batch}}$, update $r_t$ with $r_t + \lambda \frac{D_\delta(s_t)}{1 - D_\delta(s_t)}$.
11:     Use the updated $\mathcal{D}_{\text{batch}}$ and SAC to update the policy and value network parameters $\theta$ and $\psi$.
12: **end for**

---

## 3.4 Practical Algorithm

Summarizing the previous derivations, we obtain a practical reward regularization algorithm, termed as SRPO (**S**tate **R**egularized **P**olicy **O**ptimization) to leverage data with dynamics shift. We select the MAPLE [12] algorithm, which is one of the SOTA algorithms in context-based Offline RL, as the base algorithm. The detailed procedure of MAPLE+SRPO is shown in Alg. 1. After preparing the dataset in a model-based Offline RL style [40, 12], we sample a batch of data from the dataset, obtain a portion of $\rho$ states with higher rewards and add them to the dataset $\mathcal{D}_{\text{real}}$. $\mathcal{D}_{\text{fake}}$ is similarly generated by states with lower rewards (line 10). We set $\rho = 0.5$ in offline experiments with medium-expert level of data. $\rho = 0.2$ is set in all other experiments. Then a classifier discriminating data from the two datasets is trained (line 11). It estimates the logarithm of the state density ratio $\lambda \log \frac{\zeta(s)}{d_\pi(s)}$, which is added to the reward $r_t$ (line 12). $\lambda$ is regarded as a hyperparameter with values 0.1 or 0.3. The effect of $\lambda$ is investigated in Sec. 5.3. The procedure of the online algorithm CaDM [9]+SRPO is similar to MAPLE+SRPO, where the datasets $\mathcal{D}_{\text{real}}$ and $\mathcal{D}_{\text{fake}}$ are built with data from the replay buffer, rather than the offline dataset.

## 4 Theoretical Analysis

In this section, we analyze some properties of MDPs with different dynamics and provide theoretical justifications for the SRPO algorithm in Sec. 3. The notations are introduced in Sec. 2.1 and proofs can be found in Appendix A.4. We first show in Thm. 4.2 that the performance of a policy can be lower-bounded when its stationary state distribution is close to a certain optimal state distribution. In accordance with the intuition in Sec. 3.1, it is also demonstrated in Thm. 4.3 that optimal policies can induce the same stationary state distribution in different dynamics under mild assumptions. We start the analysis with the definition of homomorphous MDPs.

**Definition 4.1** (homomorphous MDPs). In an HiP-MDP $(\mathcal{S}, \mathcal{A}, \Theta, T, r, \gamma, \rho_0)$, consider hidden parameters $\theta_1, \theta_2 \in \Theta$. Let $T_i(s'|s, a) = T(s'|s, a, \theta_i), \forall (s, a, s') \in \mathcal{S} \times \mathcal{A} \times \mathcal{S}, i = 1, 2$. If $\sum_{a \in \mathcal{A}} T_1(s'|s, a) > 0 \Leftrightarrow \sum_{a \in \mathcal{A}} T_2(s'|s, a) > 0$ for all $s, s' \in \mathcal{S}$, MDPs $(\mathcal{S}, \mathcal{A}, T_1, r, \gamma, \rho_0)$ and $(\mathcal{S}, \mathcal{A}, T_2, r, \gamma, \rho_0)$ are referred to as homomorphous MDPs.

In this definition, $\sum_{a \in \mathcal{A}} T(s'|s, a) > 0$ means state $s'$ can be reached from $s$, so the equivalence of non-zero transition probabilities refers to the same reachability from $s$ to $s'$. Such condition holds in a wide range of MDPs differing only in environment parameters. For example, pendulums with different lengths can all reach the upright state from an off-center state, with longer pendulums exerting a larger force. Apart from the homomorphous property, we also require the reward and dynamics functions of MDPs to have Lipschitz properties. We assume reward function $r(s, a, s')$ w.r.t. the action $a$ is $\lambda_1$-Lipschitz and the dynamics function $T(s, a)$ w.r.t. the action $a$ is $\lambda_2$-inverse Lipschitz. Discussions on these Lipschitz properties can be found in Appendix A.5.

With these preliminaries, we first analyze the discrepancy of accumulated returns of two policies with similar stationary state distributions. The analysis is related to our SRPO algorithm in that the state regularized policy optimization formulation in Eq. (2) also constrains the learning policy to have a similar stationary state distribution with the optimal policy. Specifically, we derive a theorem as follows.

**Theorem 4.2.** *Consider two homomorphous MDPs with dynamics $T$ and $T'$. If $T' \in (T, \varepsilon_m)$, for all learning policy $\hat{\pi}$ such that $D_{\mathrm{KL}}(d_T^{\hat{\pi}}(\cdot) \| d_{T'}^*(\cdot)) \leqslant \varepsilon_s$, we have*

$$\eta_T(\hat{\pi}) \geqslant \eta_T(\pi_T^*) - \frac{\lambda_1 \lambda_2 \varepsilon_m + 2\lambda_1 + \sqrt{2} R_{\max} \sqrt{\varepsilon_s}}{1 - \gamma}. \tag{7}$$

The theorem implies that if a policy $\hat{\pi}$ has a similar stationary state distribution with the optimal policy in one MDP $M$, $\hat{\pi}$ will have a lower-bound performance guarantee in all MDPs that are homomorphous with the MDP $M$. Therefore, the learning policy can benefit from the state regularized policy optimization in Sec. 3.2.

More specifically, Eq. (7) shows that the gap in accumulated return of $\hat{\pi}$ and $\pi_T^*$ is related to the dynamics shift $\varepsilon_m$, the KL-Divergence of two stationary state distributions $\varepsilon_s$, and the effective planning horizon $\frac{1}{1-\gamma}$. With respect to the dynamics shift $\varepsilon_m$, it is related to a "uniform" constraint on the dynamics $T'$. We further show in Appendix A.4 that constraining the dynamics shift on a certain state-action pair is enough to derive Eq. (7). Unlike the dynamics shift $\varepsilon_m$ that is determined by a pre-defined RL task, the discrepancy between stationary state distributions $\varepsilon_s$ is determined by the learning policy $\hat{\pi}$ and can be optimized during training to obtain a better performance lower-bound. We also discuss in Appendix A.5 how tight Eq. (7) is in terms of the effective planning horizon $\frac{1}{1-\gamma}$, compared with some similar performance bounds.

With an additional assumption on the action gap $\Delta$ (defined in Sec. 2.1), we further demonstrate that the optimal policy of two homomorphous MDPs can have the same stationary state distribution, which verifies the intuition in Sec. 3.1.

**Theorem 4.3.** *Consider two homomorphous MDPs with dynamics $T$ and $T'$. If $T' \in (T, \varepsilon_m)$ and the action gap $\Delta$ follows $\Delta > \frac{(2-\gamma)\lambda_1\lambda_2\varepsilon_m}{1-\gamma}$, for all $s \in \mathcal{S}$ we have $d_T^*(s) = d_{T'}^*(s)$.*

The assumption is mild and holds in many scenarios. For example, in autonomous driving it can be very dangerous to deviate from the optimal policy. Such suboptimal actions have low rewards, leading to a large action gap $\Delta$. In recommendation tasks we are hardly concerned with what items we recommend (the action), as long as the recommendation outcome (the state), i.e., the users' experiences are good enough, leading to a small $\lambda_1$. The condition of large enough action gap holds in these situations.

## 5 Experiments

In this section, we conduct experiments to investigate the following questions: (1) Can SRPO leverage data with distribution shift and outperform current SOTA algorithms in the setting of HiP-MDP, in both online and offline RL? (2) How does each component of SRPO (e.g., use state regularization rather than behavior regularization) contribute to SRPO's performance? To answer question (1), we use the MuJoCo simulator [41] and generate environments with different transition functions. We train the CaDM+SRPO and the MAPLE+SRPO algorithm proposed in Sec. 3.4 and make comparative analysis with baseline algorithms. To answer question (2), we do ablation studies to examine the role of different modules in SRPO. We also examine how the discriminator $D_\delta$ works in complex environments and the effect of regularizing with state distributions in different performance levels.

### 5.1 Experiment Setup

We alter the simulator gravity to generate different dynamics in online experiments. Possible values of gravity are {1.0}, {0.7,1.0,1.3}, and {0.4,0.7,1.0,1.3,1.6} in experiments with 1, 3, and 5 kinds of different dynamics, respectively. When the simulator resets, the gravity is uniformly sampled from the set of all possible values. The number of training steps is in proportion to the number of environment parameters. Therefore, the agent has access to the same amount of training data on a

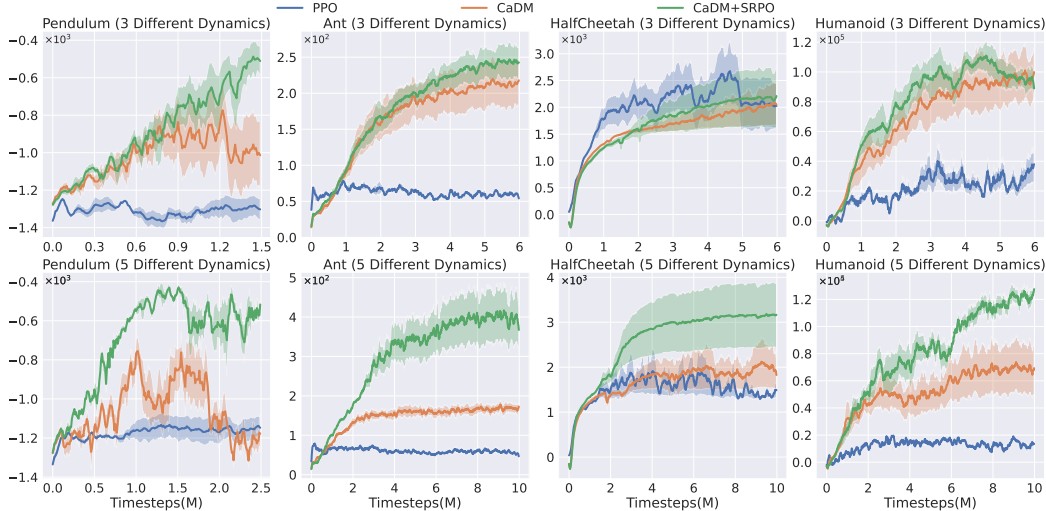

Figure 4: Results of online experiments on MuJoCo tasks. The comparison is made between CaDM+SRPO and baseline algorithms PPO, CaDM. Our CaDM+SRPO algorithm has the best overall performance in experiments with 3 and 5 different environment dynamics. The curves show the average return on 4 random seeds and the shadow areas reflect the standard deviation.

Table 1: Results of offline experiments on MuJoCo tasks. Numbers are the normalized scores according to the D4RL paper [42]. ME, M, MR and R correspond to the medium-expert, expert, medium-replay and random dataset, respectively. The evaluation is done on policies at the last iteration of training, averaged over four random seeds. The number after ± is the standard deviation. Our proposed MAPLE+SRPO algorithm has the best performance in 8 of 12 tasks and the highest overall performance.

| | CQL (Single Env) | GAIL | CQL | MOPO | MAPLE | MAPLE +DARA | MAPLE +SRPO(Ours) |
|---|---|---|---|---|---|---|---|
| Walker2d-ME | **1.11** | 0.21±0.03 | 1.03±0.10 | 0.25±0.18 | 0.55±0.21 | 0.80±0.02 | 0.66±0.08 |
| Walker2d-M | 0.79 | 0.15±0.06 | 0.78±0.01 | 0.23±0.34 | 0.82±0.01 | 0.83±0.03 | **0.84**±0.03 |
| Walker2d-MR | **0.27** | 0.00±0.00 | 0.07±0.00 | 0.00±0.00 | 0.16±0.02 | 0.17±0.01 | 0.17±0.02 |
| Walker2d-R | 0.07 | 0.00±0.00 | 0.03±0.01 | 0.00±0.00 | **0.22**±0.00 | **0.22**±0.00 | **0.22**±**0.00** |
| Hopper-ME | **0.98** | 0.04±0.01 | 0.32±0.14 | 0.01±0.00 | 0.96±0.14 | 0.96±0.06 | **0.98**±0.02 |
| Hopper-M | 0.58 | 0.00±0.00 | 0.57±0.16 | 0.01±0.00 | 0.78±0.28 | 0.40±0.05 | **1.03**±0.09 |
| Hopper-MR | 0.46 | 0.00±0.00 | 0.14±0.02 | 0.01±0.01 | 0.91±0.11 | **1.02**±0.01 | **1.02**±0.01 |
| Hopper-R | 0.11 | 0.00±0.00 | 0.11±0.00 | 0.01±0.00 | 0.13±0.00 | 0.13±0.01 | **0.32**±0.02 |
| HalfCheetah-ME | 0.62 | 0.36±0.06 | 0.03±0.04 | -0.03±0.00 | 0.50±0.06 | 0.50±0.00 | **0.63**±0.01 |
| HalfCheetah-M | 0.44 | 0.25±0.02 | 0.43±0.03 | 0.38±0.28 | 0.62±0.01 | **0.67**±0.03 | 0.63±0.01 |
| HalfCheetah-MR | 0.46 | 0.18±0.11 | 0.46±0.00 | -0.03±0.00 | 0.52±0.00 | 0.53±0.01 | **0.55**±0.00 |
| HalfCheetah-R | **0.35** | 0.14±0.02 | 0.01±0.02 | -0.03±0.00 | 0.22±0.03 | 0.21±0.00 | 0.24±0.01 |
| Average | 0.52 | 0.11 | 0.33 | 0.068 | 0.53 | 0.54 | **0.61** |

certain value of simulator gravity. We also consider the shift of medium density and body mass in offline experiments to show SRPO's robustness to different forms of dynamics shift.

To perform comparative analysis, we choose CaDM [9] and PPO [7] as baseline algorithms in online experiments. In offline experiments, DARA [5] also exploits large amount of data with dynamics shift. Its algorithm relies on Importance Sampling and will be used as a baseline method. Apart from that, we choose MOPO [40], MAPLE [12] and CQL [26] as baseline methods. More information on the setup of experiments is shown in Appendix B.1.

## 5.2 Results

**Online Experiments** The results of online experiments are shown in Fig. 4. With the context encoder and conditional policy, CaDM is able to outperform PPO in all environments. However, it fails to take advantage of the increase in the amount of data with dynamics shift. Its performance with 5 different dynamics is lower than that with 3 dynamics. In contrast, our proposed SRPO

Table 2: Results of ablation studies in offline experiments.

|  | MAPLE +SRPO | Behavior Regularizing | Random Partition | Fixed $\lambda = 0.3$ |
|---|---|---|---|---|
| Walker2d | **0.47** | 0.45 | 0.39 | 0.40 |
| Hopper | **0.83** | 0.68 | 0.56 | 0.79 |
| HalfCheetah | **0.51** | 0.50 | 0.45 | 0.40 |
| Average | **0.61** | 0.54 | 0.47 | 0.53 |

Table 3: Performance comparison of differently regularized policies.

|  | Original Hopper Env | 10x Density |
|---|---|---|
| Random | 121.3 | 44.15 |
| Medium | 2178 | 913.3 |
| Expert | 3819 | 3748 |

algorithm leads to better performance on top of CaDM in accordance with more training data. It significantly outperforms the original CaDM algorithm in environments with 5 different dynamics. The performance comparison in the Pendulum environment is also in accordance with the motivating example in Sec. 3.1. More results of online experiments are shown in Appendix B.2.

**Offline Experiments**   The results of offline experiments are shown in Tab. 1. The column of "CQL Single" refers to the evaluation score in the CQL [26] paper, where the policy is with data from a single static environment. Without the mechanism of context-based encoders, GAIL [43], CQL and MOPO [40] cannot handle data with distribution shift and show a performance drop. MAPLE [12] and MAPLE+DARA [5] only achieve marginal performance improvement with respect to CQL single. On the other hand, MAPLE+SRPO shows significant performance improvement over CQL single, which means that SRPO can efficiently leverage the additional data with dynamics shift to facilitate policy training. The MAPLE+SRPO algorithm also has a 15% higher evaluation score than MAPLE, achieving the best performance in 8 out of 12 tasks. Apart from MAPLE, the meta-RL algorithm PEARL [44] also has an context encoder for fast adaptation. We compare PEARL with PEARL+SRPO and leave the results in Appendix. B.3.

### 5.3   Analysis

**Ablations**   We conduct ablation studies in offline environments to analyze the role of each algorithm component in SRPO. The results are shown in Tab. 2. We first investigate the outcome of regularizing with state-action distribution rather than state distribution in Eq. (2). The resulting policy has a lower evaluation score on average than policies trained with the original SRPO algorithm in all environments. This is because environments with different dynamics do not have a similar optimal policy. The action distribution in the mixed dataset can be misleading when training new policies. According to Sec. 3.3, SRPO trains a classifier to discriminate states with higher values from lower values. We also train another classifier discriminating a random binary partition of states. Ablation results show a huge performance drop, which verifies the effectiveness of the classification-based surrogate mechanism. We also evaluate MAPLE+SRPO with a fixed value of the hyperparameter $\lambda$. $\lambda = 0.1$ is more suitable for Walker2d and HalfCheetah environments, while in the Hopper environment $\lambda = 0.3$ is better. This is in accordance with previous analysis that Hopper agents can benefit more from regularizing with the stationary state distribution.

**Effectiveness of Discriminators**   We first train a discriminator $D_\delta$ according to Alg. 1. Then a set of states is sampled from the D4RL [42] dataset and classified into two sets according to the output of $D_\delta$. The average values of the two sets of states are compared in Fig. 5. As shown in the figure, states with higher $D_\delta$ outputs also have higher values in all three environments. It means that the trained discriminator $D_\delta$ can successfully identify states with high values from those with low values. Therefore, its output can be a good surrogate for the density ratio $\frac{\zeta(s)}{d_\pi(s)}$ in Sec. 3.

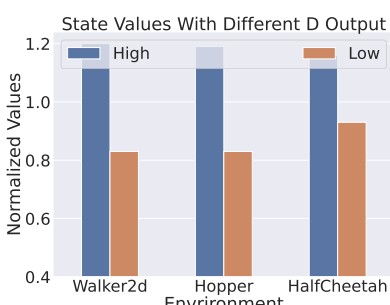

Figure 5: Comparison of values on states with high and low output of the discriminator $D$.

**Effectiveness of the Regularization**   We also study the effect of policy regularization with different performance levels of stationary state distributions. Random, medium and expert policies in the original Hopper environment are

used to estimate the stationary state distributions, which regularize the learning policies in a new environment with different dynamics. The results are shown in Tab. 3, where the expert policy is the most effective in regularizing. This verifies the practice in Sec. 3.2 and the theoretical analysis.

## 6    Conclusion and Discussion

In this work, we focus on the problem of leveraging data with dynamics shift to efficiently train RL agents. Based on the intuition that optimal policies can lead to similar stationary state distributions, we give a constrained optimization formulation that regards the state distribution as a regularizer. After discussions on a sample-based surrogate, we propose the SRPO algorithm which can be an add-on module to context-based algorithms and improve their sample efficiency. The resulting CaDM+SRPO and MAPLE+SRPO algorithms show superior performance when learning on data sampled from environments with different dynamics. Theoretical analyses are also given to analyze some properties of MDPs with different dynamics. They provide justifications for the intuition of the dynamics-invariant state distribution, as well as the constrained policy optimization formulation.

**Limitations and Future work**    The theoretical analyses of this work requires the assumption of homomorphous MDPs, i.e., the same state reachability in different MDPs. It would be interesting to discuss whether similar conclusions on the stationary state distribution can be derived without such assumption.

**Acknowledgements**    We thank Wanqi Xue and Yanchen Deng for helpful discussions. This research is supported by the National Research Foundation, Singapore under its Industry Alignment Fund – Pre-positioning (IAF-PP) Funding Initiative and Ministry of Education, Singapore, under its Academic Research Fund Tier 1 (RG13/22). Any opinions, findings and conclusions or recommendations expressed in this material are those of the author(s) and do not reflect the views of National Research Foundation, Singapore.

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
