# A Additional Derivations and Proofs

## A.1 Derivations of the Lagrangian

We start from the optimization problem:

$$\max_{\pi} \; \mathbb{E}_{s_t, a_t \sim \tau_\pi} \sum_{t=0}^{\infty} \gamma^t r\left(s_t, a_t\right) \tag{1}$$
$$\text{s.t.} \quad D_{\mathrm{KL}}\left(d_\pi(\cdot)\|\zeta(\cdot)\right) < \varepsilon_m.$$

The KL-Divergence term can be transformed as:

$$
\begin{aligned}
D_{\mathrm{KL}}\left(d_\pi(\cdot)\|\zeta(\cdot)\right) &= -\mathbb{E}_{s \sim d_\pi(s)}\left[\log \zeta(s) - \log d_\pi(s)\right] \\
&= -\int d_\pi(s)\left[\log \zeta(s) - \log d_\pi(s)\right] ds \\
&= -\int (1-\gamma)\sum_{t=0}^{\infty} \gamma^t p\left(s_t = s\right)\left[\log \zeta(s) - \log d_\pi(s)\right] ds \\
&= -(1-\gamma)\sum_{t=0}^{\infty}\int \gamma^t p\left(s_t = s\right)\left[\log \zeta(s) - \log d_\pi(s)\right] ds \\
&= -(1-\gamma)\sum_{t=0}^{\infty}\mathbb{E}_{s_t \sim \tau}\left[\gamma^t\left(\log \zeta(s_t) - \log d_\pi(s_t)\right)\right] \\
&= -(1-\gamma)\mathbb{E}_{s_t \sim \tau}\sum_{t=0}^{\infty}\gamma^t\left(\log \zeta(s_t) - \log d_\pi(s_t)\right).
\end{aligned}
\tag{2}
$$

So the constraint can be written as

$$\mathbb{E}_{s_t \sim \tau}\sum_{t=0}^{\infty}\left[\gamma^t\left(\log d_\pi(s_t) - \log \zeta(s_t)\right)\right] - \frac{\varepsilon_m}{1-\gamma} < 0. \tag{3}$$

The optimization problem can be written as the following standard form

$$\min_{\pi} \; \mathbb{E}_{s_t, a_t \sim \tau}\sum_{t=0}^{\infty} -\gamma^t r\left(s_t, a_t\right) \tag{4}$$
$$\text{s.t.} \quad \mathbb{E}_{s_t \sim \tau}\sum_{t=0}^{\infty}\left[\gamma^t\left(\log d_\pi(s_t) - \log \zeta(s_t)\right)\right] - \frac{\varepsilon_m}{1-\gamma} < 0.$$

So the Lagrangian $L$ is

$$L = -\mathbb{E}_{s_t, a_t \sim \tau}\left[\sum_{t=0}^{\infty}\gamma^t\left(r(s_t, a_t) + \lambda \log \zeta(s_t) - \lambda \log d_\pi(s_t)\right)\right] - \frac{\lambda \varepsilon_m}{1-\gamma}. \tag{5}$$

## A.2 Derivations of the Forward and Backward Probabilities

The backward probability can be written as:

$$
\begin{aligned}
\beta_t(s_t) &= \int_{\mathcal{S}} p(\mathcal{O}_{t:\infty}|s_t, s_{t+1}, \pi)p(s_{t+1}|s_t)ds_{t+1} \\
&= \int_{\mathcal{S}} p(\mathcal{O}_t|s_t, \pi)p(\mathcal{O}_{t+1:\infty}|s_{t+1}, \pi)p(s_{t+1}|s_t)ds_{t+1} \\
&= \int_{\mathcal{S}} \max_{a_t}\exp(\gamma^t r(s_t, a_t))\beta_{t+1}(s_{t+1})p(s_{t+1}|s_t)ds_{t+1}.
\end{aligned}
\tag{6}
$$

Taking logarithm on both sizes, we have

$$\log \beta_t(s_t) = \log \mathbb{E}_{s_{t+1}}\max_{a_t}\exp(\gamma^t r(s_t, a_t) + \log \beta_{t+1}(s_{t+1})). \tag{7}$$

Let $W(s_t) = \log \beta_t(s_t)$, we get

$$W(s_t) = \log \mathbb{E}_{s_{t+1}} \exp \left[ \max_{a_t} \gamma^t r(s_t, a_t) + W(s_{t+1}) \right]. \tag{8}$$

According to [1], $W_t$ is a soft version of the traditional value function $V_t$. As the Soft Actor-Critic [2] has become the base algorithm in many scenarios, $\beta_t$ is closely related to the value function learned during training, which is often in its soft version. The forward probability $\alpha_t(s_t) = p\left(\mathcal{O}_{0:t-1} \mid s_t, \pi\right)$ is the probability of trajectory from timestep $0$ to $t-1$ being optimal given the state $s_t$. Such probability is hard to model as the transition from $s_{t-1}$ to $s_t$ is related to the actual policy $\pi$ as well as the environment dynamics. Therefore, we do not take $\alpha_t(s_t)$ into account when dividing training data to train the classifier.

## A.3 Discussions on the Surrogate for the Density Ratio

According to some Off-Policy RL algorithms [3, 4], the idea of training a classifier $D(s)$ as a data-based surrogate of the density ratio $\frac{\zeta(s)}{d_\pi(s)}$ can also be derived from a theorem related to f-divergence (lemma 1 in [4]). Such derivation is essentially the same with our GAN-based proposition. Technically, these algorithms also divides the training data into two parts and train a classifier, which is later used to generate probabilities for prioritized sampling. Our SRPO algorithm proposes a different criterion to divide the training data, and train a classifier used in reward augmentation.

## A.4 Proofs to Theorems in Sec. 4

We first introduce the following lemma which is essential in proving the two theorems in Sec. 4.

**Lemma A.1.** *Consider two homomorphous MDPs with dynamics $T$ and $T'$. Assuming $T' \in (T, \varepsilon_m)$, the reward function w.r.t. the action is $\lambda_1$-Lipschitz and the dynamics function w.r.t. the action is $\lambda_2$-inverse Lipschitz, we have*

$$|V_T^*(s) - V_{T'}^*(s)| \leqslant \frac{\lambda_1 \lambda_2 \varepsilon_m}{1 - \gamma} \tag{9}$$

*for all $s \in \mathcal{S}$.*

*Proof.* Recall the optimal value function under dynamics $T$ follows

$$V_T^*(s) = \max_a \ r(s, a, T(s, a)) + \gamma V_T^*(T(s, a)). \tag{10}$$

Without the loss of generality, we assume $V_T^*(s) \geqslant V_{T'}^*(s)$ on a certain state $s$. Define $a_T^* = \pi_T^*(s)$ and $\hat{a}$ such that $T'(s, \hat{a}) = T(s, a_T^*) = s'$. Then we have

$$|T(s, a_T^*) - T(s, \hat{a})| = |T'(s, \hat{a}) - T(s, \hat{a})| \leqslant \varepsilon_m, \tag{11}$$

and

$$|r(s, a_T^*, s') - r(s, \hat{a}, s')| \leqslant \lambda_1 |a_T^* - \hat{a}| \leqslant \lambda_1 \lambda_2 \varepsilon_m. \tag{12}$$

Therefore for all $s \in \mathcal{S}$,

$$\begin{aligned}
|V_T^*(s) - V_{T'}^*(s)| &= V_T^*(s) - V_{T'}^*(s) \\
&= r(s, a_T^*, s') + \gamma V_T^*(s') - \max_a \ [r(s, a, T'(s, a)) + \gamma V_{T'}^*(T'(s, a))] \\
&\leqslant r(s, a_T^*, s') + \gamma V_T^*(s') - r(s, \hat{a}, s') - \gamma V_{T'}^*(s') \\
&\leqslant \lambda_1 \lambda_2 \varepsilon_m + \gamma |V_T^*(s') - V_{T'}^*(s')| \\
&\leqslant \lambda_1 \lambda_2 \varepsilon_m + \gamma \lambda_1 \lambda_2 \varepsilon_m + \gamma^2 |V_T^*(s'') - V_{T'}^*(s'')| \\
&\leqslant \cdots \\
&\leqslant \frac{\lambda_1 \lambda_2 \varepsilon_m}{1 - \gamma},
\end{aligned} \tag{13}$$

which concludes the proof. $\qquad\square$

This lemma shows the discrepancy upper bound between the optimal state value functions in two homomorphous MDPs. We then apply it to prove the second theorem in Sec. 4.

**Theorem A.2** (Restatement of Thm. 4.3). *Following the assumptions in Lem. A.1, if the action gap $\Delta$ follows $\Delta > \frac{(2-\gamma)\lambda_1\lambda_2\varepsilon_m}{1-\gamma}$, for all $s \in \mathcal{S}$ we have $d_T^*(s) = d_{T'}^*(s)$.*

*Proof.* Recall that the definition of action gap is $\Delta = \min\limits_{\theta \in \Theta} \min\limits_{s \in \mathcal{S}} \min\limits_{a \neq \pi_T^*(s)} V_{T_\theta}^*(s) - Q_{T_\theta}^*(s, a)$. Therefore, we have

$$
\begin{aligned}
V_T^*(s) &\geqslant Q_T^*(s, a) + \Delta \\
&> Q_T^*(s, a) + \frac{(2-\gamma)\lambda_1\lambda_2\varepsilon_m}{1-\gamma}
\end{aligned}
\tag{14}
$$

for all $(s, a) \in \mathcal{S} \times \mathcal{A}$ if $a \neq \pi_T^*(s)$. The same property holds for the transition function $T'$. We first show the state transition probability derived from $\pi_T^*$ and $\pi_{T'}^*$ is the same: $p_T(\cdot|s, \pi_T^*) = p_{T'}(\cdot|s, \pi_{T'}^*)$, $\forall s \in \mathcal{S}$. Without the loss of generality, let $V_T^*(s) \geqslant V_{T'}^*(s)(*)$. Let

$$
\begin{aligned}
\bar{a} &= \arg\max_a r(s, a, T(s, a)) + \gamma V_T^*(T(s, a)) \\
a' &= \arg\max_a r(s, a, T'(s, a)) + \gamma V_{T'}^*(T'(s, a)) \\
T'(s, \tilde{a}) &= T(s, \bar{a}) = \bar{s}, \ T'(s, a') = s'.
\end{aligned}
\tag{15}
$$

According to Eq. (12), $\|\tilde{a} - \bar{a}\| \leqslant \lambda_1\lambda_2\varepsilon_m$. Supposing $\bar{s} \neq s'(**)$, we have $\tilde{a} \neq a' = \pi_{T'}^*(s)$. So

$$
V_{T'}^*(s) > Q_{T'}^*(s, \tilde{a}) + \frac{(2-\gamma)\lambda_1\lambda_2\varepsilon_m}{1-\gamma}.
\tag{16}
$$

Meanwhile,

$$
\begin{aligned}
Q_{T'}^*(s, \tilde{a}) &= r(s, \tilde{a}, \bar{s}) + \gamma V_{T'}^*(\bar{s}) \\
&\geqslant r(s, \bar{a}, \bar{s}) + \gamma V_{T'}^*(\bar{s}) - \lambda_1\lambda_2\varepsilon_m \\
&\geqslant r(s, \bar{a}, \bar{s}) + \gamma V_T^*(\bar{s}) - \lambda_1\lambda_2\varepsilon_m - \frac{\gamma\lambda_1\lambda_2\varepsilon_m}{1-\gamma} \\
&= V_T^*(s) - \frac{(2-\gamma)\lambda_1\lambda_2\varepsilon_m}{1-\gamma}
\end{aligned}
\tag{17}
$$

Combining Eq. (16) and Eq. (17), we get $V_{T'}^*(s) > V_T^*(s)$, which contradicts with Eq. $(*)$. It means that the assumption $(**)$ is not correct, so $\bar{s} = s'$.

We then show that $d_T^*(s) = d_{T'}^*(s)$ for all $s \in \mathcal{S}$:

$$
\begin{aligned}
&\|p_T(s_t = \cdot|\pi_T^*) - p_{T'}(s_t = \cdot|\pi_{T'}^*)\|_\infty \\
&= \left\|\sum_{s'} p_T(\cdot|s', \pi_T^*)p_T(s_{t-1} = s'|\pi_T^*) - p_{T'}(\cdot|s', \pi_{T'}^*)p_{T'}(s_{t-1} = s'|\pi_{T'}^*)\right\|_\infty \\
&= \left\|\sum_{s'} p_T(\cdot|s', \pi_T^*)\left[p_T(s_{t-1} = s'|\pi_T^*) - p_{T'}(s_{t-1} = s'|\pi_{T'}^*)\right]\right\|_\infty \\
&\leqslant \left\|\sum_{s'} p_T(\cdot|s', \pi_T^*)\|p_T(s_{t-1} = \cdot|\pi_T^*) - p_{T'}(s_{t-1} = \cdot|\pi_{T'}^*)\|_\infty\right\|_\infty \\
&= \left\|\|p_T(s_{t-1} = \cdot|\pi_T^*) - p_{T'}(s_{t-1} = \cdot|\pi_{T'}^*)\|_\infty \sum_{s'} p_T(\cdot|s', \pi_T^*)\right\|_\infty \\
&= \|p_T(s_{t-1} = \cdot|\pi_T^*) - p_{T'}(s_{t-1} = \cdot|\pi_{T'}^*)\|_\infty \\
&\leqslant \cdots \\
&\leqslant \|p_T(s_0 = \cdot|\pi_T^*) - p_{T'}(s_0 = \cdot|\pi_{T'}^*)\|_\infty \\
&= 0.
\end{aligned}
\tag{18}
$$

Therefore, for all $s \in \mathcal{S}$, we have $p_T(s_t = s|\pi_T^*) = p_{T'}(s_t = s|\pi_{T'}^*)$. So

$$
|d_T^*(s) - d_{T'}^*(s)| = \left|\sum_{t=0}^{\infty} p_T(s_t = s|\pi_T^*) - p_{T'}(s_t = s|\pi_{T'}^*)\right| = 0
\tag{19}
$$

for all $s \in \mathcal{S}$, which concludes the proof. $\qquad\square$

Before proving the first theorem in Sec. 4, we introduce a lemma that incorporates the 1-Wasserstein distance between the policies. It also considers a reference policy that has the same stationary state distribution with the optimal policy in the other dynamics. Such policy exists thanks to the homomorphous property of the MDPs.

**Lemma A.3.** *Following the assumptions in Lem. A.1, for all policy $\hat{\pi}$ such that $d_T^{\hat{\pi}}(s) = d_{T'}^*(s)$ for all $s \in \mathcal{S}$ and $\max_s W_1\left(\hat{\pi}(\cdot|s), \pi_{T'}^*(\cdot|s)\right) \leqslant \epsilon_\pi$, we have*

$$|\eta_T(\pi_T^*) - \eta_T(\hat{\pi})| \leqslant \frac{\lambda_1 \lambda_2 \varepsilon_m + \lambda_1 \varepsilon_\pi}{1 - \gamma}, \tag{20}$$

*where $W_1(\hat{\pi}(\cdot|s), \pi_{T'}^*(\cdot|s))$ is the 1-Wasserstein distance between two policies.*

*Proof.* First, $|\eta_T(\pi_T^*) - \eta_{T'}(\pi_{T'}^*)|$ can be bounded with Thm. A.1:

$$
\begin{aligned}
|\eta_T(\pi_T^*) - \eta_{T'}(\pi_{T'}^*)| &= |\mathbb{E}_{s \in \rho_0} V_T^*(s) - \mathbb{E}_{s \in \rho_0} V_{T'}^*(s)| \\
&\leqslant \frac{\lambda_1 \lambda_2 \varepsilon_m}{1 - \gamma}.
\end{aligned}
\tag{21}
$$

We then try to bound $|\eta_T(\hat{\pi}) - \eta_{T'}(\pi_{T'}^*)|$. We first define the state-action stationary distributions $D_1(s, a) = d_T^{\hat{\pi}}(s)\hat{\pi}(a|s)$ and $D_2(s, a) = d_{T'}^*(s)\pi_{T'}^*(a|s)$. The accumulated return can be written as

$$
\begin{aligned}
\eta_T(\hat{\pi}) &= \frac{1}{1 - \gamma} \mathbb{E}_{s,a,s' \sim D_1}\left[r(s, a, s')\right] \\
\eta_{T'}(\pi_{T'}^*) &= \frac{1}{1 - \gamma} \mathbb{E}_{s,a,s' \sim D_2}\left[r(s, a, s')\right]
\end{aligned}
\tag{22}
$$

We start from the Lipschitz property of the reward function:

$$|r(s, a_1, s') - r(s, a_2, s')| \leqslant \lambda_1 \|a_1 - a_2\|_1. \tag{23}$$

Taking expectation w.r.t. $d_{T'}^*(\cdot)$ on both sides, we get

$$\mathbb{E}_{s \sim d_{T'}^*}|r(s, a_1, s') - r(s, a_2, s')| \leqslant \mathbb{E}_{s \sim d_{T'}^*} \lambda_1 \|a_1 - a_2\|_1. \tag{24}$$

Letting $\mu(A_1, A_2|s)$ be any joint distribution with marginals $\hat{\pi}$ and $\pi_{T'}^*$ conditioned on $s$. Taking expectation w.r.t. $\mu$ on both sides, we get

$$
\begin{aligned}
|\mathbb{E}_{s,a \sim D_1} r(s, a, s') - \mathbb{E}_{s,a \sim D_2} r(s, a, s')| &\leqslant \mathbb{E}_{s \sim d_{T'}^*} \mathbb{E}_{a_1,a_2 \sim \mu}|r(s, a_1, s') - r(s, a_2, s')| \\
&\leqslant \lambda_1 \mathbb{E}_{s \sim d_{T'}^*} E_\mu \|a_1 - a_2\|_1 \\
&\leqslant \max_s \lambda_1 E_\mu \|a_1 - a_2\|_1.
\end{aligned}
\tag{25}
$$

Eq. (25) holds for all joint distribution $\mu$, so it also holds for $\bar{\mu} = \arg\min_\mu \lambda_1 E_\mu \|a_1 - a_2\|_1$, leading to the 1-Wasserstein distance:

$$|\mathbb{E}_{s,a \sim D_1} r(s, a, s') - \mathbb{E}_{s,a \sim D_2} r(s, a, s')| \leqslant \max_s \lambda_1 W_1(\hat{\pi}(\cdot|s), \pi_{T'}^*(\cdot|s)). \tag{26}$$

According to Eq. (22), we have

$$|\eta_T(\hat{\pi}) - \eta_{T'}(\pi_{T'}^*)| \leqslant \frac{\lambda_1 \varepsilon_m}{1 - \gamma}. \tag{27}$$

Applying the triangle inequality, we get

$$
\begin{aligned}
|\eta_T(\pi_T^*) - \eta_T(\hat{\pi})| &\leqslant |\eta_T(\pi_T^*) - \eta_{T'}(\pi_{T'}^*)| + |\eta_T(\hat{\pi}) - \eta_{T'}(\pi_{T'}^*)| \\
&\leqslant \frac{\lambda_1 \lambda_2 \varepsilon_m + \lambda_1 \varepsilon_\pi}{1 - \gamma},
\end{aligned}
\tag{28}
$$

which concludes the proof.

We then use this lemma to prove the first theory in Sec. 4. $\qquad\square$

Table 1: $\lambda_1, \lambda_2, R_{\max}$ in practical environments.

| Environment | Action-related Reward | $\lambda_1$ | $\lambda_2$ | $R_{\max}$ |
|---|---|---|---|---|
| CartPole-v0 | 0 | 0 | 1.42 | 1.00 |
| InvertedPendulum-v2 | 0 | 0 | 8.58 | 1.00 |
| Swimmer-v2 | $-0.0001|a|_2^2$ | 0.0001 | 2.59 | 0.36 |
| HalfCheetah-v2 | $-0.1|a|_2^2$ | 0.1 | 1.01 | 4.80 |
| Hopper-v2 | $-0.001|a|_2^2$ | 0.001 | 3.45 | 3.80 |
| Walker2d-v2 | $-0.001|a|_2^2$ | 0.001 | 4.70 | $\geqslant 4$ |
| Ant-v2 | $-0.5|a|_2^2$ | 0.5 | 0.69 | 6.0 |
| Humanoid-v2 | $-0.1|a|_2^2$ | 0.1 | 0.03 | $\geqslant 8$ |

**Theorem A.4** (Restatement of Thm. 4.2). *Consider two homomorphous MDPs with dynamics $T$ and $T'$. If $T' \in (T, \varepsilon_m)$, for all learning policy $\hat{\pi}$ such that $D_{\mathrm{KL}}(d_T^{\hat{\pi}}(\cdot)\|d_{T'}^*(\cdot)) \leqslant \varepsilon_s$, we have*

$$\eta_T(\hat{\pi}) \geqslant \eta_T(\pi_T^*) - \frac{\lambda_1\lambda_2\varepsilon_m + 2\lambda_1 + \sqrt{2}R_{\max}\sqrt{\varepsilon_s}}{1-\gamma}. \tag{29}$$

*Proof.* In two homomorphous MDPs with dynamics $T$ and $T'$, there exists a policy $\tilde{\pi}$ such that $d_T^{\tilde{\pi}}(\cdot) = d_{T'}^*(\cdot)$. According to Lem. A.3, we have

$$|\eta_T(\pi_T^*) - \eta_T(\tilde{\pi})| \leqslant \frac{\lambda_1\lambda_2\varepsilon_m + \lambda_1\varepsilon_\pi}{1-\gamma} \leqslant \frac{\lambda_1\lambda_2\varepsilon_m + 2\lambda_1}{1-\gamma}, \tag{30}$$

where the second inequality is obtained as the actions are bounded to $[-1, 1]$. The scaling is multiplied by the Lipschitz coefficient which tends to small, so it will make little influence to the bound. On the other hand, policies $\hat{\pi}$ and $\tilde{\pi}$ have a similar state discrepancy: $D_{\mathrm{KL}}(d_T^{\hat{\pi}}(\cdot)\|d_T^{\tilde{\pi}}(\cdot)) \leqslant \varepsilon_s$. Therefore, their performance gap can be bounded according to results in imitation learning (Lem. 6 in [5]):

$$|\eta_T(\hat{\pi}) - \eta_T(\tilde{\pi})| \leqslant \frac{\sqrt{2}R_{\max}\sqrt{\varepsilon_s}}{1-\gamma}. \tag{31}$$

Merging Eq. (30) and Eq. (31), we obtain

$$\begin{aligned}
\eta_T(\pi_T^*) - \eta_T(\hat{\pi}) &\leqslant |\eta_T(\hat{\pi}) - \eta_T(\pi_T^*)| \\
&\leqslant |\eta_T(\pi_T^*) - \eta_T(\tilde{\pi})| + |\eta_T(\hat{\pi}) - \eta_T(\tilde{\pi})| \\
&\leqslant \frac{\lambda_1\lambda_2\varepsilon_m + 2\lambda_1 + \sqrt{2}R_{\max}\sqrt{\varepsilon_s}}{1-\gamma}.
\end{aligned} \tag{32}$$

$\square$

### A.5 Discussions on the Theoretical Analysis

**The Lipschitz Assumptions in Sec. 4** Regarding the reward functions, the Lipschitz property implies that if $s$ and $s'$ keeps unchanged, the deviation of the reward $r$ will be no larger than $\lambda_1$ times the deviation of the action $a$. Therefore, the Lipstchiz coefficient $\lambda_1$ is solely related to action-related terms in the reward function. It is important to note that different actions exist given the same $s$ and $s'$ since we may compute the reward function in different dynamics. Considering the dynamics functions, the Lipschitz property indicates that if the current state $s$ remains unchanged and the actions differ by : $|a_1 - a_2| \geqslant \varepsilon$, the next states will exhibit a significant difference: $|s_1' - s_2'| \geqslant \frac{\varepsilon}{\lambda_2}$.

In Tab. 1, we list the action-related terms of the reward functions for various RL evaluation environments, along with the corresponding values of $\lambda_1$ derived from these terms. Additionally, we sample 50,000 $(s, a, s')$ tuples from the replay buffer, slightly modify the action, and observe how the resulting next state $s'$ changes. The replay buffer contains trajectories collected during different training phases and should be diverse enough to cover most possible trajectories. This empirical analysis allows us to calculate $\lambda_2$ in practice. As indicated in the table, the action-related terms in reward functions exhibit reasonably small coefficients in all environments, leading to small $\lambda_1$ values. Combined with medium values of $\lambda_2$, it can be inferred that Lipschitz terms, including $\lambda_1$ and $\lambda_1\lambda_2$, will remain small in practical scenarios and will not dominate the error term in Eq. 7. Also, the action gap assumption in Thm. 4.3 (line 258-259) is not strong and holds in many situations.

Table 2: Detailed results of ablation studies in offline experiments.

| | MAPLE +SRPO | Behavior Regularization | Random Partition | Fixed $\lambda = 0.1$ | Fixed $\lambda = 0.3$ |
|---|---|---|---|---|---|
| Walker2d-medium-expert | 0.66±0.08 | **0.70**±0.18 | 0.42±0.16 | 0.66±0.08 | 0.38±0.16 |
| Walker2d-medium | **0.84**±0.03 | 0.71±0.02 | 0.79±0.00 | 0.72±0.13 | **0.84**±0.03 |
| Walker2d-medium-replay | **0.17**±0.02 | 0.16±0.01 | 0.14±0.01 | **0.17**±0.02 | 0.16±0.01 |
| Walker2d-random | **0.22**±0.00 | **0.22**±0.00 | **0.22**±0.00 | **0.22**±0.00 | **0.22**±000 |
| Hopper-medium-expert | **0.98**±0.02 | 0.85±0.25 | 0.46±0.14 | **0.98**±0.02 | 0.86±0.18 |
| Hopper-medium | **1.03**±0.01 | 0.78±0.26 | 0.76±0.21 | 0.53±0.13 | **1.03**±0.01 |
| Hopper-medium-replay | **1.02**±0.01 | 0.94±0.04 | 0.91±0.08 | 1.02±0.01 | 0.93±0.03 |
| Hopper-random | **0.32**±0.02 | 0.13±0.01 | 0.12±0.01 | 0.13±0.01 | **0.32**±0.02 |
| Halfcheetah-medium-expert | 0.63±0.01 | **0.65**±0.01 | 0.44±0.18 | 0.63±0.01 | 0.52±0.00 |
| Halfcheetah-medium | **0.63**±0.01 | 0.60±0.00 | 0.62±0.02 | 0.61±0.02 | **0.63**±0.01 |
| Halfcheetah-medium-replay | **0.55**±0.00 | 0.54±0.00 | 0.54±0.01 | **0.55**±0.00 | 0.24±0.01 |
| Halfcheetah-random | **0.24**±0.01 | 0.21±0.03 | 0.20±0.01 | **0.24**±0.01 | 0.23±0.01 |
| Average | **0.61** | 0.54 | 0.47 | 0.54 | 0.53 |

**Failure Cases**   Although the assumptions are weak and hold in many situations, there are certain scenarios that these assumptions do not hold and the performance of SRPO can not be guaranteed. For example, in maze environments with different obstacle layout, the requirement of homomorphous MDPs is violated. There are also cases where the Lipstchitz coefficients $\lambda_1, \lambda_2$ can be large, such as stock markets with very high transaction feeds.

**The Assumption on Dynamics Discrepancy**   We mentioned in Sec. 4 that one of the assumptions to prove the theorems is that $T \in (T', \varepsilon_m)$. In fact, this is a simplification of the actual requirement, which is weaker than the uniform bound of dynamics shift. According to Eq. (11), for any state $s$ we only require $T(s, a)$ and $T'(s, a)$ to be close on one specific action $\hat{a}$ such that $T'(s, \hat{a}) = T(s, a_T^*) = s'$. This is a point-wise bound on dynamics shift and is comparable to assumptions in previous analysis [6].

**The Tightness of Eq. (29)**   Eq. (29) has a similar form with the Eq. (1) in Thm. 4.1 of [7], where the return discrepancy $|\eta_T(\pi_T^*) - \eta_T(\hat{\pi})|$ is also bounded by differences in the policy distribution and transition functions, with an order of two in the effective horizon (i.e. with a coefficient $\frac{1}{(1-\gamma)^2}$). By introducing some assumptions and constraining the policy to have the same stationary state distribution, we obtain a tighter discrepancy bound with an order of one in the effective horizon (i.e. with a coefficient $\frac{1}{1-\gamma}$).

# B   Experiment Details

## B.1   Setup

To generate environments with different transition functions, We alter the xml file of the MuJoCo simulator and change its environment parameters. In online experiments, we build our code based on the Github repository[1] of CaDM [8]. Some customized MuJoCo environments are defined in this repository. They have different reward functions with the original environments. We keep these modifications to make our online results comparable with the original CaDM algorithm. In offline experiments, we build our code based on the Github repository[2]. The offline datasets are generated by concatenating the data sampled in the original MuJoCo simulator, as well as simulators whose gravity and medium density are altered. In both online and offline experiments, the evaluation is done in online static environments with all possible values of environment parameters. The average of these evaluation results is reported.

---

[1] https://github.com/younggyoseo/CaDM/tree/master
[2] https://github.com/polixir/OfflineRL

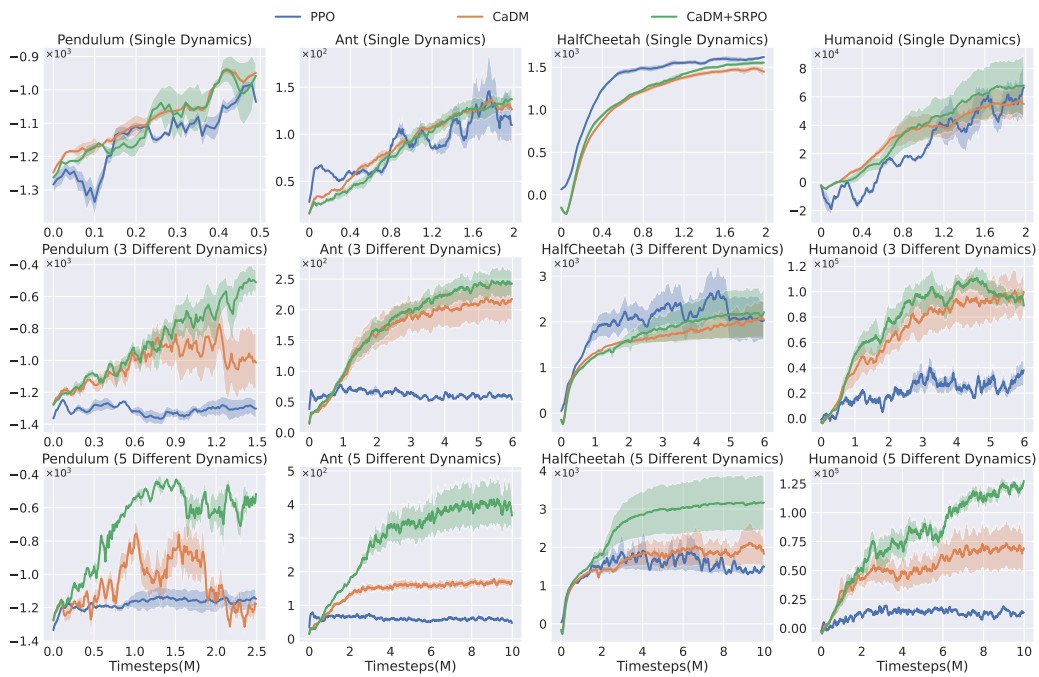

Figure 1: Detailed results of online experiments on MuJoCo tasks. In environments with single dynamics, three algorithms have a similar performance.

Table 3: Results of offline experiments with a small dataset.

| | MOPO | MAPLE | MAPLE+ DARA | MAPLE+ SRPO (ours) |
|---|---|---|---|---|
| Walker2d-medium | $0.21 \pm 0.13$ | $0.45 \pm 0.18$ | $0.74 \pm 0.12$ | $\mathbf{0.79} \pm 0.04$ |
| Walker2d-medium-expert | $0.14 \pm 0.06$ | $0.26 \pm 0.01$ | $0.38 \pm 0.03$ | $\mathbf{0.61} \pm 0.11$ |
| Hopper-medium | $0.01 \pm 0.00$ | $0.42 \pm 0.36$ | $0.36 \pm 0.06$ | $\mathbf{0.51} \pm 0.14$ |
| Hopper-medium-expert | $0.01 \pm 0.00$ | $0.33 \pm 0.09$ | $0.16 \pm 0.04$ | $\mathbf{0.40} \pm 0.06$ |
| HalfCheetah-medium | $0.10 \pm 0.01$ | $0.50 \pm 0.06$ | $0.37 \pm 0.01$ | $\mathbf{0.55} \pm 0.03$ |
| HalfCheetah-medium-expert | $-0.03 \pm 0.00$ | $0.35 \pm 0.01$ | $\mathbf{0.63} \pm 0.03$ | $0.62 \pm 0.19$ |
| Average | $0.07$ | $0.39$ | $0.44$ | $\mathbf{0.58}$ |

## B.2    Additional Results

We show full results of online experiments on MuJoCo tasks in Fig. 1. Experiments on environments with single dynamics are included. These experiments are equivalent to those on static static environments. PPO, CaDM and CaDM+SRPO have a similar performance in these tasks. Full results of ablation studies are shown in Tab. 2. We also reduce the amount of offline data to 1/3 and perform additional experiments. The results are shown in Tab. 3. MAPLE+SRPO can still achieve better performance than baseline algorithms. It improves the performance by 31% over MAPLE+DARA and 49% over MAPLE. These evidences indicate that SRPO indeed enables efficient data reuse, which is in accordance with statements in the introduction part.

## B.3    Additional Analysis

To provide an additional demonstrating example to the intuition in Sec. 3.1, we train two policies in the Pendulum environment with 0.5 and 2 times of the original frictions and then visualize state and action densities. The results in Fig. 2 are similar to the experiments altering the environment gravity. We observe similar state distributions and different peaks in action distributions.

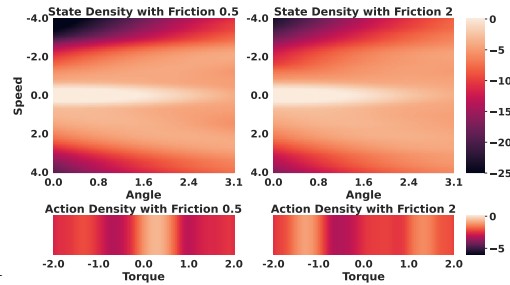

Figure 2: Visualization of state and action densities in data sampled from the Inverted Pendulum environment with 0.5 and 2 times of the original friction. Under both frictions, the state distribution has high density with low pendulum speed and small pendulum angle. Meanwhile, the action distribution has different peaks in density under different frictions.

With respect to different Offline RL tasks, MAPLE+SRPO gains the highest rise in the Hopper environment and outperform all baseline methods in all of the 4 tasks. In the Walker2d and HalfCheetah environments, however, MAPLE+SRPO only outperforms in half of the tasks. Such difference results from the existence of multiple optimal policies which lead to different stationary state distributions [9, 10]. For example, the agent in the Walker2d environment has many ways to swing its arms to keep balance. When the policy pattern in the offline dataset is different from the learning policy, its stationary state distribution may not be a good regularizer. The Hopper agent has a fewer degree of freedom compared with the other two, so the policy benefits more from regularizing with SRPO.