# OpenReview forum: "State Regularized Policy Optimization on Data with Dynamics Shift"
_NeurIPS.cc/2023/Conference — NeurIPS 2023 poster_

### Official Review · Reviewer_6EPu · 2023-06-17

**Soundness:** 4 excellent
**Presentation:** 4 excellent
**Contribution:** 3 good
**Rating:** 7
**Confidence:** 5

**Summary:**

The paper proposes a context-aware method called SRPO (State Regularized Policy Optimization) to address the issue of training RL algorithms on data with dynamics shift, i.e., with different underlying environment dynamics. The authors reveal and exploit the property of similar stationary state distributions in environments with similar structures but different dynamics. The learned stationary state distribution in SRPO is used to regularize policies trained in new environments, leading to more efficient data reuse and cross-domain ability. The paper also provides a lower-bound performance guarantee on policies regularized by the optimal state distribution in other dynamics. The experimental results show that SRPO can be a flexible add-on design that makes context-based RL algorithms far more data efficient and significantly improve their overall performance.

**Strengths:**

+ The paper is well-structured and easy to follow, with relatively rigorous experimental validations and inspiring theoretical analyses.
+ The proposed SRPO algorithm is a context-aware method that tackles the challenge of training RL algorithms on data exhibiting diverse dynamics shifts, a very general problem in many real-world scenarios.
+ The authors have identified and effectively leveraged the observation that environments with similar structures but different dynamics possess similar optimal stationary state distributions. This discovery facilitates efficient data reuse and demonstrates improved performance in both online and offline settings, as evidenced by the experimental results.
+ The paper makes a valuable theoretical contribution by providing a lower-bound performance guarantee on policies regularized by the optimal state distribution in alternative dynamics.

**Weaknesses:**

+ Please refer to the subsequent question section.

**Questions:**

##### Method:
+ While the state regularized objective in SRPO is indeed well-motivated and illustrated by examples in the Inverted Pendulum environment, it would benefit from additional theoretical evidence to firmly establish the assertion that environments with similar structures but different dynamics inherently share similar optimal stationary state distributions. It should be noted that such a claim may not always hold true in some environments, and further theoretical investigation is expected to determine the generalizability of this finding.
+ Though the sample-based surrogate mechanism for state density ratio estimation is relatively well-founded, there is still some confusion regarding the inability of $D_{\text{fake}}$ with lower state values to capture the stationary state distribution $d_\pi$.  It remains a challenge to handle $D_{\text{fake}}$ effectively, as it tends to underestimate the quality of the training policy and may even introduce a mixture of $d_\pi$ and $d_{\pi_\beta}$  induced by the behavior policy in static datasets in offline settings.
##### Experimental results:
+ DARC and DARA objective is derived from constrained optimization with dynamics regularization, instead of state regularization in SRPO, which both decouple the regularization term with policy that might vary in environments with dynamics discrepancies. It would be beneficial and interesting to provide more extensive analyses or explanations about these two types of regularization and demonstrate in what situations SRPO can perform larger advantage and why SRPO can work with more cases. It might also accout for the results of offline experiments that "MAPLE+DARA" offers very competing performance against "MAPLE+SRPO" on halfcheetah and walker tasks.
+ Further elaboration is needed to support the statement that SRPO enables efficient data reuse, specifically regarding reduced data consumption from source and target domains.
+ If there are any specific design considerations aimed at stabilizing the training process, it would be beneficial to include a detailed list of implementation details related to discriminator training.

##### Typos:
- line 139 in Appendix: repeated "static".

**Limitations:**

The authors have adequately and precisely addressed the limitations in this work.

---

> ### Author Rebuttal · Authors · 2023-08-08
>
> Thank you for your valuable feedback! We provide discussions on your concerns as follows.
>
> Q: **The paper would benefit from additional theoretical analysis on the intuition of similar optimal state distributions.**
>
> A: Thm. 4.3 (lines 258-259) of our paper provides evidence for the existence of such similar state distributions. We restate the theorem here:
>
> Consider two homomorphous MDPs with dynamics $T$ and $T^{\prime}$. If $T^{\prime} \in\left(T, \varepsilon_m\right)$ and the action gap $\Delta$ follows $\Delta>\frac{(2-\gamma) \lambda_1 \lambda_2 \varepsilon_m}{1-\gamma}$, for all $s \in \mathcal{S}$ we have $d_T^*(s)=d_{T^{\prime}}^*(s)$.
>
> In this theorem, $T$ and $T^{\prime}$ refer to two distinct dynamics with similar structures. $d_T^*(s)$ and $d_{T^{\prime}}^*(s)$ are optimal state distributions in these two environment dynamics. With the assumption of large enough action gap $\Delta$ (which is mild as analyzed in line 260-265), it can be proved that $d_T^*(s)$ will be the same with $d_{T^{\prime}}^*(s)$. This is in accordance with our initial intuition of similar optimal state distributions and the design of the SRPO algorithm. These discussions will also be added in the revision of our paper.
>
> Q: **It would be beneficial and interesting to provide more extensive analyses or explanations about the two types of regularization (DARA style and SRPO style) and demonstrate in what situations SRPO can perform larger advantage and why SRPO can work with more cases.**
>
> A:  Thanks for pointing it out. We will add the following analysis to the experiment part in the revision. According to Tab. 2 of our paper, SRPO emerges as the top-performing algorithm across all environments. However, in the Walker2d and HalfCheetah environments SRPO only exhibits a relatively small margin of improvement over the DARA style of algorithm (behavior regularizing). This observation can be attributed to the dynamics functions of Walker2d and HalfCheetah tasks being less affected by the altered environment parameters. In other words, these two tasks are easier to solve. Such task property is in turn substantiated by the performance of baseline algorithms without context encoders, such as GAIL and MOPO, which manage to achieve non-zero performance scores in these environments.
>
> In contrast, in environments characterized by more significant dynamics deviations, such as Hopper, SRPO demonstrates a more pronounced margin of improvement over baseline algorithms. This heightened performance can be attributed to SRPO's focus on a more general property — the intuition of similar optimal state distributions — enabling it to effectively address the challenges posed by environments with larger dynamics deviations.
>
>
> Q: **Further elaboration is needed to support the statement that SRPO enables efficient data reuse.**
>
> A: Thanks for pointing it out. We will add the following elaborations to the experiment analysis in the revision. As for *online experiments*, Fig. 4 of our paper demonstrates that in 7 out of 8 experiment settings, SRPO consistently outperforms baseline algorithms throughout the training process, even with very few training steps. This observation highlights SRPO's ability to achieve good performance with limited training data in online settings, making it a more data-efficient approach.
>
> In *offline experiments*, we reduce the amount of offline data to 1/3 and perform additional experiments. The results are shown in the following table, where M and ME refer to Medium and Medium-Expert performance levels of offline data. MAPLE+SRPO can still achieve better performance than baseline algorithms. **MAPLE+SRPO improves the performance by 31\% over MAPLE+DARA and 49\% over MAPLE**. These evidences indicate that SRPO indeed enables efficient data reuse, which is in accordance with statements in the introduction part.
>
> |                |      MOPO       |      MAPLE      |    MAPLE+DARA     | MAPLE+SRPO (ours)  |
> | -------------- | :-------------: | :-------------: | :---------------: | :----------------: |
> | Walker2d-M     | 0.21 $\pm$ 0.13 | 0.45 $\pm$ 0.18 |  0.74$\pm$ 0.12   | **0.79**$\pm$0.04  |
> | Walker2d-ME    |  0.14$\pm$0.06  | 0.26 $\pm$ 0.01 |  0.38$\pm$ 0.03   | **0.61**$\pm$ 0.11 |
> | Hopper-M       |  0.01$\pm$0.00  |  0.42$\pm$0.36  |   0.36$\pm$0.06   | **0.51**$\pm$0.14  |
> | Hopper-ME      |  0.01$\pm$0.00  |  0.33$\pm$0.09  |   0.16$\pm$0.04   | **0.40**$\pm$0.06  |
> | HalfCheetah-M  |  0.10$\pm$0.01  |  0.50$\pm$0.06  |   0.37$\pm$0.01   | **0.55**$\pm$0.03  |
> | HalfCheetah-ME | -0.03$\pm$0.00  |  0.35$\pm$0.01  | **0.63**$\pm$0.03 |   0.62$\pm$0.19    |
> | Average        |      0.07       |      0.39       |       0.44        |        **0.58**        |
>
> Q: **If there are any specific design considerations aimed at stabilizing the training process, it would be beneficial to include a detailed list of implementation details related to discriminator training.**
>
> A: We would like to emphasize that the training of the discriminator in our paper is inherently different from that in generative adversarial networks (GANs). Unlike GANs, which are notoriously challenging to stabilize during training, our approach does not involve a generator module. Consequently, the discriminator in our study serves as a simple classifier, leading to quick convergence and enhanced training stability, as shown in Fig. 2 of the one-page pdf. Therefore, it is unnecessary to take specific design considerations to stabilize the training of the discriminator.

---

> > ### Comment · Reviewer_6EPu · 2023-08-14
> >
> > Thanks for the rebuttal. These supplementary explanations and experiments can definitely strengthen the authors' empirical claims in this paper, while I still have some questions and advice:
> >
> > > **"It would be beneficial and interesting to provide more extensive analyses or explanations about the two types of regularization (DARA style and SRPO style) and demonstrate in what situations SRPO can perform larger advantage and why SRPO can work with more cases."**
> >
> > I agree with the explanation regarding the relative ease of solving the HalfCheetah and Walker tasks, and their resilience to dynamics shifts. However, it would greatly enhance the paper if the authors could offer additional substantiation and analyses to elucidate why augmenting rewards with the **logarithm of stationary state distribution ratio** - as implemented in SRPO - outperforms the alternative approach of augmenting rewards with the **logarithm of dynamics ratio**, as observed in DARC-style algorithms such as DARA. Both lines of works hold theoretical validity, and further insights could provide a deeper understanding. My guessing could be that approximating the ratio of stationary state distributions lends itself to a more convergent and stable estimation process with neural networks.
> >
> > > **"As for online experiments, Fig. 4 of our paper demonstrates that in 7 out of 8 experiment settings, SRPO consistently outperforms baseline algorithms throughout the training process, even with very few training steps."**
> >
> > I see that SRPO doesn't show notably better performance in the initial training steps. In many experiments depicted in Figure 4, CaDM and CaDM+SRPO exhibit similar average returns in the first 2 million steps. The distinction becomes clearer when considering the long-term or asymptotic performance, where SRPO seems to have an advantage.
> >
> > > **Failure cases.**
> >
> > I recommend that the authors could add the potential failure cases to the appendix for further inspirations to this community since the theoretical assumptions might not always hold for complex real world cases though it can coverage most of the common ones.

---

> > > ### Author Response · Authors · 2023-08-15
> > > **Replying to Reviewer 6EPu**
> > >
> > > Thank you for your timely response and detailed questions/suggestions! We provide discussions on your concerns as follows.
> > >
> > > Q1: **Comparison between SRPO and DARC style of reward augmentation**
> > >
> > > A1: We concur with the idea that approximating the ratio of stationary state distributions can be more stable and converge quickly. The reason behind this is that in practice, DARA employs two distinct discriminators that estimates the ratio $\frac{q\left(\text {source} \mid \mathbf{s}, \mathbf{a}, \mathbf{s}'\right)}{q\left(\text{target} \mid \mathbf{s}, \mathbf{a}, \mathbf{s}'\right)}$  and $\frac{q(\text {source} \mid \mathbf{s}, \mathbf{a})}{q(\operatorname{target} \mid \mathbf{s}, \mathbf{a})}$ separately. The discriminators will receive $(s,a,s^\prime)$ and $(s,a)$ as input, respectively. In contrast, SRPO employs a single discriminator that estimates the ratio $\frac{p(\text{Real}|\textbf{s})}{p(\text{Fake}|\textbf{s})}$ . It solely requires the state $s$ as input. The additional inputs of action and next state can **increase the complexity of the discriminator in DARA and lead to a slower convergence**. Also, the joint distributions over $(s,a,s')$ and $(s,a)$, which appear in the denominator of the ratios, are more prone to have values close to zero. This can make the reward augmentation unstable and have high variance. Motivated by the intuition of similar optimal state distribution, SRPO focus on a more fundamental property in environments with dynamics shift and provides a simpler and more efficient approach of reward augmentation. This can explain why SRPO can outperform DARA in most of the offline RL tasks.
> > >
> > > To empirically substantiate the aforementioned discussions, we select the task of Walker-medium-expert and show the loss of the different discriminators of SRPO and DARA on the 100th epoch of training. Notably, the discriminator in SRPO demonstrates a smaller loss. The complete loss curve will be added in the revision.
> > >
> > > | Discriminator Input | $(s)$ (SRPO) | $(s, a)$ (DARA) | $\left(s, a, s^{\prime}\right)$ (DARA) |
> > > | :--- | :---: | :---: | :---: |
> > > | Loss on the 100th epoch | 0.84 | 1.03 | 1.12 |
> > >
> > > Q2:  **SRPO doesn't show notably better performance in the initial training steps**
> > >
> > > A2:  In online experiments, SRPO works as an add-on module to the CaDM algorithm. Apart from reward augmentation, all other modules and hyperparameters remain unchanged. Therefore, in the initial training steps the performance of SRPO+CaDM can be constrained by the exploration setup of the base algorithm CaDM, which is itself a value-based maximum-entropy reinforcement learning algorithm based on SAC. Since the training steps are the total interaction steps with all environments, in the first 2M steps the agent only interacts with each environment for 667k steps when there are 3 different dynamics, and for 400k steps with 5 different dynamics. The online policy can have high entropy and be highly stochastic with such few training steps. As a result, SRPO+CaDM may not show notably better performance in the initial training steps.
> > >
> > > Q3: **The theoretical assumptions might not always hold for complex real world cases though it can coverage most of the common ones.**
> > >
> > > A3: Potential failure cases will indeed be helpful for further exploration of the similar assumptions. We will add a failure example of maze environments with different obstacle layout, which violates the requirement of homomorphous MDPs (lines 223-256). We will also add an example task of stock market trading with very high transaction fees, which tend to have a high Lipstchitzs coefficient  (lines 232-234).
> > >
> > >
> > > We will add the aforementioned discussions in the revision of our paper. We sincerely appreciate the questions and suggestions you provide in the two responses. We believe that solving the questions and adding the relevant discussions will greatly enhance our paper.

---

> > > > ### Comment · Reviewer_6EPu · 2023-08-15
> > > >
> > > > Thanks for the reply. The additional explanations and analyses address my concerns properly and would be a strong addition to this paper, so I would keep voting for an acceptance. Wish you all the best with your publication.

---

> > > > > ### Author Response · Authors · 2023-08-15
> > > > > **Replying to Reviewer 6EPu**
> > > > >
> > > > > We thank the reviewer again for the valuable feedback. As we have addressed the concerns pointed out during the official review, would you like to kindly consider raising your score accordingly?

---

### Official Review · Reviewer_ZrWb · 2023-07-06

**Soundness:** 3 good
**Presentation:** 3 good
**Contribution:** 3 good
**Rating:** 6
**Confidence:** 3

**Summary:**

The paper considers leveraging existing data in RL but with dynamics shifts. The proposed method augments the reward with a density ratio estimation term that encourage the policy to recover the optimal state distribution under the distribution shift. The theory motivates the algorithm design by showing that under certain assumptions, optimal policies under similar dynamics have close stationary state distribution. Finally, the experiments show that the proposed method indeed works well in practice in both online and offline setting with dynamics shift.



**Strengths:**

1. The paper is well-written and easy to follow.

2. The proposed method can be easily added to some previous methods.

3. The paper provides motivating theory results to back up the design of the practical algorithm.

4. The design of the density ratio estimation design is well motivated in Section 3.3.

5. The paper provides good coverage of the experimental setting (both online and offline and includes several environments in the mujoco benchmark). The result is pretty good compared with previous baselines.

**Weaknesses:**

1. The theory section (section 4) only provides some motivation for the algorithm design, instead of actually analyzing the algorithm.

2. The Lipchitzness of the reward function and dynamics still seems pretty strong even with section A.5: it should argument not only the Lipchitzness condition holds, and also the coefficients should be reasonably small.



**Questions:**

N/A

---

> ### Author Rebuttal · Authors · 2023-08-09
>
> Thank you for your valuable feedback! We provide discussions on your concerns as follows.
>
> Q: **The theory section (section 4) only provides some motivation for the algorithm design, instead of actually analyzing the algorithm.**
>
> A: The SRPO algorithm works by augmenting the reward function. Such augmentation changes the original MDP and it is usually hard to analyze the sample complexity or convergence rate of the new algorithm. These analyses are also absent in similar reward augmentation algorithms, such as [DARA](https://openreview.net/forum?id=9SDQB3b68K), [DARC](https://openreview.net/forum?id=eqBwg3AcIAK) and [H2O](https://arxiv.org/abs/2206.13464). Meanwhile, the experiments results can serve as empricial analyses of the SRPO algorithm.
>
> Q: **The Lipchitzness of the reward function and dynamics still seems pretty strong even with section A.5: it should argument not only the Lipchitzness condition holds, and also the coefficients should be reasonably small.**
>
> A: Thanks for pointing it out. Please refer to the general response where we discuss how $\lambda_1$ and $\lambda_2$ should be reasonably small and do not dominate the error term in practice. We will add the discussions in the revision.

---

> > ### Comment · Reviewer_ZrWb · 2023-08-17
> > **Response**
> >
> > I acknowledge the authors' rebuttal and I remain my rating towards acceptance.

---

> > > ### Author Response · Authors · 2023-08-18
> > > **Replying to Reviewer ZrWb**
> > >
> > > Thank you for your positive feedback and recognition! We are grateful for your time and efforts in reviewing our paper and giving constructive comments.

---

### Official Review · Reviewer_PXH4 · 2023-07-06

**Soundness:** 2 fair
**Presentation:** 2 fair
**Contribution:** 2 fair
**Rating:** 4
**Confidence:** 3

**Summary:**

This paper presents an approach to RL/policy learning in the case where the dynamics parameters are potentially time-varying. The approach relies on the view that the optimal state distribution is similar across (small) variations in the underlying parameters, subject to assumptions. Thus, the authors propose an approach that constrains the distance between the realized state distribution under the trained policy and that of the optimal policy. Because the optimal policy state distribution is unknown, the authors propose to approximate the ratio of state distributions via discriminative methods, which in turn rely on an RL-as-inference formulation that leverages the value function.

**Strengths:**

- The problem is reasonable, and guaranteeing (in some sense) reasonable RL performance with time-varying parameters is an important step toward practical robustness.
- The method is reasonable and original.
- The experimental evaluation is reasonably comprehensive.

**Weaknesses:**

- The underlying motivation of similar state distributions is questionable. The similarity of these distributions is said to arise when the dynamics are similar (in this case, as reflected by (T, eps) notation) and when the reward functions and dynamics functions are Lipschitz. Then, the reward difference is approximately bounded by the products of the Lipschitz constants and the eps difference between dynamics functions. The authors claim these assumptions are mild---on the contrary, these conditions are quite strong and the resulting bounds are weak. In general, the idea of targeting the same state distribution doesn't seem reasonable for many real world systems, especially since a common use-case of RL is to handle non-smooth dynamics.
- The approach is quite complex. The addition of a value-dependent discriminator is a substantial addition to standard RL workflows, and this added complexity will likely be a barrier to adoption. The authors should consider investigating simplified approaches---perhaps motivated by the same goals---that are more likely to see adoption.
- Most importantly, the difference MAPLE + SRPO over MAPLE is extremely minor. Indeed, across many tasks the difference in performance is not statistically significant. In general, the combination of the high added complexity and the small resulting performance difference makes it difficult to recommend acceptance of the paper.

**Questions:**

- Is it possible to avoid the training of new discriminator while still obtaining the benefits of state distribution regularization?
- Is it possible to generalize analysis beyond smooth systems?

**Limitations:**

The conditions stated in the paper (discussed above) are described as mild. The conditions are not mild, and the limitations of these assumptions should be discussed in more detail.

---

> ### Author Rebuttal · Authors · 2023-08-08
>
> Thank you for your valuable feedback! We provide discussions on your concerns as follows.
>
> Q: **The similarity of similar state distributions is said to arise when the dynamics are similar and when the reward functions and dynamics functions are Lipschitz. These conditions are quite strong and the resulting bounds are weak.**
>
> A: We would like to respectfully counter with the argument of strong conditions and weak bounds. We restate the main performance bound here:
> $$
> \eta_T(\hat{\pi}) \geqslant \eta_T\left(\pi_T^*\right)-\frac{\lambda_1 \lambda_2 \varepsilon_m+2 \lambda_1+\sqrt{2} R_{\max } \sqrt{\varepsilon_s}}{1-\gamma},
> $$
>
> where $\lambda_1$ and $\lambda_2$ are Lipstchitz coefficients, $\varepsilon_s$ is the error of state regularization $D_{\mathrm{KL}}\left(d_T^{\hat{\pi}}(\cdot) \| d_{T^{\prime}}^*(\cdot)\right)$, and $R_{\max }$ is the maximum possible reward. As clarified in the general response, $\lambda_1 \lambda_2$ and $\lambda_1$ are more than 20 times smaller than $R_{\max }$, making the term with $R_{\max }$ dominate the error bound. Additionally, in similar performance bounds such as Thm. 1 in TRPO and Thm. 4.1 in MBPO, $R_{\max }$ is multiplied by squared planning horizon $\frac{1}{(1-\gamma)^2}$. Thanks to the assumptions on Lipstchitsz conditions, our bound is stronger than these bounds in that $R_{\max }$ is multiplied by $\frac{1}{1-\gamma}$.
>
> Q: **The underlying motivation of similar state distributions is questionable. In general, the idea of targeting the same state distribution doesn't seem reasonable for many real world systems, especially since a common use-case of RL is to handle non-smooth dynamics.**
>
> A: We would like to emphasize that the assumption was made on the **$\lambda_2$-inverse** Lipstchitz property of the dynamics function, rather than the more commonly used $\lambda$-Lipstchitz property. This crucial distinction ensures that our assumption does not overly constrain practical tasks to have smooth dynamics, and the underlying motivation remains valid even in tasks with non-smooth dynamics.
> To illustrate this point, consider the standard cliff walking task as an example. In this scenario, the optimal state trajectory involves the agent walking alongside the cliff towards the goal position. Even with a small change in action, the agent may fall off the cliff, resulting in a significant variation in the next state. Hence, this task exhibits non-smooth dynamics. Despite the non-smooth dynamics, the optimal state distribution remains consistent across different dynamics. It will always have high-density areas near the cliff.
>
> Q: **The approach is quite complex. The addition of a value-dependent discriminator is a substantial addition to standard RL workflows, and this added complexity will likely be a barrier to adoption.**
>
> A: Introducing an additional discriminator is a common and widely adopted practice in RL community, as demonstrated in various studies such as [LFIW](https://arxiv.org/abs/2006.13169), [ReMERT](https://arxiv.org/abs/2105.07253), [DARC](https://openreview.net/forum?id=eqBwg3AcIAK), and [DARA](https://arxiv.org/abs/2203.06662). In our paper, we incorporate the discriminator module into three different RL algorithms: CaDM, MAPLE and PEARL (in appendix). Remarkably, all three algorithms exhibit excellent performance, further highlighting the versatility and adaptability of the additional discriminator across different areas of RL. With respect to time complexity, we show the running time comparison of MAPLE, MAPLE+DARA (the baseline method), and MAPLE+SRPO (our method) in the following table. SRPO only needs about 15% of the extra training time and is more efficient than DARA.
>
> |  | Training Time (h) |  |  |
> | :---| :---: | :---: | :---:|
> |  | MAPLE | MAPLE+DARA | MAPLE+SRPO |
> | Hopper-v2 | 5.47 | 6.32 | 6.14 |
> | Walker2d-v2 | 7.20 | 8.48 | 8.25 |
> | HalfCheetah-v2 | 7.05 | 8.32 | 8.19 |
>
> It is also crucial to note that the training procedure of the discriminator in our paper is fundamentally different from that in generative adversarial networks (GANs). Unlike GANs, which are notoriously challenging to stabilize during training, our approach does not involve a generator module. Consequently, the discriminator in our study serves as a simple classifier, leading to quick convergence and enhanced training stability, as shown in Fig. 2 of the one-page pdf. We will add these discussions and statistics in the revision.
>
> Q: **The difference of MAPLE + SRPO over MAPLE is extremely minor.**
>
> A: MAPLE+SRPO and MAPLE are algorithms designed for the domain of offline RL in environments with dynamics shift. In this setting, performance improvements from different algorithms are generally limited due to the data-collecting policy (or behavior policy) being directly trained in the target environments and thus not affected by dynamics shift.
>
>  As shown in the following table from [H2O](https://arxiv.org/abs/2206.13464), [DARC](https://openreview.net/forum?id=eqBwg3AcIAK) achieved ~10% improvement over the baseline algorithm CQL. On top of DARC, H2O achieved ~11.5% performance improvement. Considering our approach, the MAPLE+SRPO algorithm demonstrates a 15% increase in performance compared to MAPLE and a 13% improvement over MAPLE+DARA. These results indicate a significant and reasonable enhancement achieved by the MAPLE+SRPO algorithm in the context of offline RL with dynamics shift.
> |  | CQL | DARC | H2O |
> | :--- | :--- | :--- | :--- |
> |Average Performance  | 5196 | 5715 | 6373 |
> | Improvement over baseline  | N/A | 9.98\% | 11.5\% |
>
> On the other hand, in online RL the data-collecting policy is the same as the policy that is being trained.  When a policy cannot adapt to environments with dynamics shifts, it will struggle to collect high-quality data to train and have a very poor performance. The second row of Fig. 4 in our paper demonstrates the large performance gap of CaDM+SRPO with other baseline algorithms in online experiments.

---

> ### Author Response · Authors · 2023-08-18
> **Replying to Reviewer PXH4**
>
> We sincerely value your dedicated guidance in helping us enhance our work. We are eager to ascertain whether our responses adequately address your primary concerns, particularly in relation to the underlying assumptions, the complexity of the approach, and the performance comparisons. We would be grateful for the opportunity to provide any needed further feedback.

---

### Official Review · Reviewer_mfzT · 2023-07-10

**Soundness:** 3 good
**Presentation:** 4 excellent
**Contribution:** 3 good
**Rating:** 6
**Confidence:** 4

**Summary:**

The paper aims to improve sample efficiency for RL algorithms trained in environments with dynamics shift. The core motivation is that, for different environments and their corresponding optimal policies, the state visitation distribution could be similar. Building upon this motivation, the authors propose a constrained policy optimization (CPO) that encourages the learned policy to have a similar stationary state distribution given by the optimal state distribution. For theoretical analysis, the authors propose homomorphous MDPs, which require two MDPs to reach the next state with the same support given the same current state and action. The result is that, given the reward and dynamics functions of MDPs having Lipschitz properties, if a policy $\hat{\pi}$ has a similar stationary state distribution with the optimal policy in one MDP $M$, $\hat{\pi}$ will have a lower-bound performance guarantee in all MDPs that are homomorphous with the MDP $M$. Experimental results show that SRPO outperforms existing state-of-the-art methods in both online and offline RL settings.

**Strengths:**

The motivation that, for similar MDPs, their optimal policies should have similar stationary state distributions is very intuitive and makes a lot of sense. This motivation is well supported by both theoretical analysis and experimental results.

**Weaknesses:**

Please see the questions section.

**Questions:**

1. In equation 5, we use $R_\max$ directly. What if $R_\max$ is unknown? I assume we can estimate $R_\max$ simply by running several rollouts, or set $R_\max$ manually from domain knowledge. And what if the maximum reward we can obtain is different (or even has a different scale) for different states? E.g., we have $R_\max$ as 100, but for state $s_1$ the maximum reward is only 0, and for state $s_2$ the maximum reward is 100. Given equation 5, we always subtract $r(s_t, a_t)$ by 100 for both $s_1$ and $s_2$, but this seems unfair for $s_t$.
2. Line 159, there is no D.3 in the appendix.
3. In some cases, the assumption that the optimal policy for similar MDPs has similar stationary state distributions might not hold. For example , driving a car (with a fixed power level) on roads with different slopes, and the reward function is the velocity of the car. Apparently, the maximum velocity on different roads can be different, thus affecting the stationary state distribution. We can find more examples like this, only if the stationary state distribution for the optimal policy is strongly correlated with the MDP parameter. We do not expect an algorithm to be applicable in all cases, but I think we should clearly state the scope of this algorithm.
4. In both Figure 1 and the experiment section, we use a discrete set of MDPs. Several papers mentioned in line 104 "MDPs with Different Dynamics" indeed consider the case of continuous sets of MDPs, like VariBAD. For VariBAD, the generalization between different MDPs is carried out by the continuity of policy w.r.t. MDP dynamics. And it seems inappropriate to conclude "data with different dynamics are used in an ad hoc manner. The large amount of data with different dynamics are discarded during training, leading to poor sample efficiency (line 38)". Data collected from one MDP is helpful for other MDPs that are close to it. Can we 1) run experiments on continuous sets of MDPs, 2) make the discussion in line 37 more accurate? And if time permits, consider the VariBAD baseline.

**Limitations:**

The authors did not address limitations or potential negative societal impact of their work. No further improvement should be made.

---

> ### Author Rebuttal · Authors · 2023-08-08
>
> Thank you for your valuable feedback! We provide discussions on your concerns as follows.
>
> Q: **In equation 5, we use Rmax directly. What if Rmax is unknown or different across states?**
>
> A: In Eq. 5, $R_\max$ serve as a normalization factor to guarantee the probability $p(\mathcal{O}\_t|s\_t)$ is positive. In [Sergey 2018](https://arxiv.org/abs/1805.00909), $R_\max$ is simply set to 0 without loss of generality. Also, what we really care about is the optimality of a state trajectory. When estimating the probability of a trajectory being optimal, single-state optimal probabilities are multiplied together. Different $R_\max$ across states do not pose an issue when considering a whole state trajectory.
>
> Q: **Line 159, there is no D.3 in the appendix.**
>
> A: Sorry for this blunder. D.3 in Line 159 refers to Appendix C.3. We will change it in the revision.
>
> Q: **In some cases, the assumption might not hold. Consider the car driving scenario.**
>
> A: Thanks for pointing it out. We will add the following discussion in the revision. In the theory part (Section 4) of the paper, we provide the condition under which using optimal state distributions as regularizers will be helpful for policy training (Thm. 4.2). So theoretically SRPO is expected to work effectively in tasks following such condition. The condition contains the homomorphous property of MDP and two Lipstchitz properties, where the Lipstchitz properties hold in a large variety of environments, as elucidated in the general response.
>
> In the car driving scenario the intuition of similar optimal state distribution does not hold, in that the homomorphous property is violated. Such property requires the reachability from $s$ to $s'$ is the same in the source and target environments. In the example of car driving, the power levels are fixed. Therefore, cars on slopes that are steeper can not accelerate as fast as cars on other slopes, leading to different reachability. Nevertheless, in most RL tasks the homomorphous property holds in at least states that are frequently visited, as demonstrated by the superior performance of SRPO in both online and offline experiments.
>
> Q: **Can we 1) run experiments on continuous sets of MDPs, 2) make the discussion in line 37 more accurate, 3) consider the VariBAD baseline?**
>
> A: We will change the discussion in line 37 to: "The generalization ability of context encoders relies on the expressive power of neural networks. However, neural networks are prone to overfit and behave poorly when doing extrapolation".  Also note that SRPO can provide context encoders with a generalization ability **orthorgorical** to that of neural networks. Both generalization effect can work together. Therefore, SRPO will also work well in a continuous set of MDP. Indeed, the comparison of PEARL+MAPLE and PEARL in the appendix is made on a continous set of Walker2d environments, where PEARL+MAPLE has better performance.
>
> Regarding the VariBAD baseline, we conducted additional experiments in Pendulum, Ant, HalfCheetah and Humanoid environments, each with 5 different dynamics. The results are shown in Fig. 1 of the one-page pdf in the general response. **VariBAD exhibits a better performance compared with CaDM and PPO baselines, but can not outperform our CaDM+SRPO algorithm**. We would also like to mention that as a general add-on module, SRPO can also work together with VariBAD, resulting in the VariBAD+SRPO algorithm.

---

> > ### Comment · Reviewer_mfzT · 2023-08-16
> >
> > Thank you for submitting your rebuttal. The authors have made a commendable effort to address and answer my concerns, which is greatly appreciated.
> >
> > In relation to my third question, I kindly suggest that the authors consider incorporating additional intuitions into the main body of the paper rather than relying solely on theoretical statements around Line 242. Additionally, I would like to highlight that another reviewer has also raised the issue of failure cases, which I concur with. It would be highly beneficial if the authors could address this aspect as well.

---

> > > ### Author Response · Authors · 2023-08-16
> > > **Replying to Reviewer mfzT**
> > >
> > > Thank you for your positive feedback and recognition! We are grateful for your time and efforts in reviewing our paper and giving constructive comments. We will add the additional intuitions in the main body of our paper and add the failure cases in the appendix.

---

> > > ### Author Response · Authors · 2023-08-18
> > > **Replying to Reviewer mfzT**
> > >
> > > We thank the reviewer again for the valuable feedback. As we have addressed the concerns pointed out during the official review, especially the additional baselines, would you like to kindly consider raising your score accordingly?

---

### Author Rebuttal · Authors · 2023-08-08

We extend our sincere gratitude to all the reviewers for providing insightful comments. Both reviewer PXH4 and ZrWb mentioned the absence of discussions on the Lipstchitz coefficients in the assumptions. In this common response, we will address this issue and discuss how Lipstchitz coefficients can be small and negligible in practical scenarios. These discussions will be incorporated into the revised version of our paper.

We first restate our assumptions on the Lipschitz properties: The reward function $r(s, a, s^{\prime})$ w.r.t. the action $a$ is $\lambda_1$-Lipschitz and the dynamics function $T(s, a)$ w.r.t. the action $a$ is $\lambda_2$-inverse Lipschitz. According to Eq. 7 (line 241-242), two terms related to Lipschitz properties are considered: the multiplication $\lambda_1 \lambda_2$ and $\lambda_1$ itself.

Regarding the reward functions, the Lipschitz property implies that if $s$ and $s^{\prime}$ keeps unchanged, the deviation of the reward $r$ will be no larger than $\lambda_1$ times the deviation of the action $a$. Therefore, the Lipstchiz coefficient $\lambda_1$ is solely related to action-related terms in the reward function. It is important to note that different actions exist given the same $s$ and $s^{\prime}$ since we may compute the reward function in different dynamics. Considering the dynamics functions, the Lipschitz property indicates that if the current state $s$ remains unchanged and the actions differ by : $\|a_1-a_2\| \geqslant \varepsilon$, the next states will exhibit a significant difference: $\|s_1^{\prime}-s_2^{\prime}\| \geqslant \frac{\varepsilon}{\lambda_2}$.

In the following table, we list the action-related terms of the reward functions for various RL evaluation environments, along with the corresponding values of $\lambda_1$ derived from these terms. Additionally, we sample 50,000 (s,a,s') tuples from the replay buffer, slightly modify the action, and observe how the resulting next state $s^{\prime}$ changes. The replay buffer contains trajectories collected during different training phases, and should be diverse enough to cover most possible trajectories. This empirical analysis allows us to calculate $\lambda_2$ in practice and demonstrate it in the table.

| Environment | Action-related Reward | $\lambda_1$ | $\lambda_2$ | $R_{\max }$ |
| :--- | :--- | :--- | :--- | :--- |
| CartPole-v0 | 0 | 0 | 1.42 | 1.00 |
| InvertedPendulum-v2 | 0 | 0 | 8.58 | 1.00 |
| Swimmer-v2 | $-0.0001\|a\|_2^2$ | 0.0001 | 2.59 | 0.36 |
| HalfCheetah-v2 | $-0.1\|a\|_2^2$ | 0.1 | 1.01 | 4.80 |
| Hopper-v2 | $-0.001\|a\|_2^2$ | 0.001 | 3.45 | 3.80 |
| Walker2d-v2 | $-0.001\|a\|_2^2$ | 0.001 | 4.70 | $\geqslant 4$ |
| Ant-v2 | $-0.5\|a\|_2^2$ | 0.5 | 0.69 | 6.0 |
| Humanoid-v2 | $-0.1\|a\|_2^2$ | 0.1 | 0.03 | $\geqslant 8$ |

As indicated in the table, the action-related terms in reward functions exhibit reasonably small coefficients across all environments, leading to small $\lambda_1$ values. Combined with medium values of $\lambda_2$, it can be inferred that Lipstchitz terms, including $\lambda_1$ and $\lambda_1\lambda_2$,  will remain small in practical scenarios and will not dominate the error term in Eq. 7. Also, the action gap assumption in Thm. 4.3 (line 258-259) is not strong and holds in many situations.

---

### Decision · Program_Chairs · 2023-09-21

**Decision:**

Accept (poster)

**Comment:**

The paper proposes a context-aware method called SRPO (State Regularized Policy Optimization) to address the issue of training RL algorithms on data with dynamics shifts. Paper is well-written and easy to follow, both theoretical analysis (lower-bound performance guarantees) and experiments are intuitive and strong. The setting studied is also general enough to be applicable to many real-world scenarios. Methods can also be applied to both online and offline RL settings.